# Chemical characterization and source identification of PM$_{2.5}$ at multiple sites in the Beijing-Tianjin-Hebei region, China

**Xiaojuan Huang[1,2], Zirui Liu[1,3*], Jingyun Liu[1], Bo Hu[1], Tianxue Wen[1], Guiqian Tang[1], Junke Zhang[1], Fangkun Wu[1], Dongsheng Ji[1], Lili Wang[1], Yuesi Wang[1,3*]**

[1]State Key Laboratory of Atmospheric Boundary Layer Physics and Atmospheric Chemistry (LAPC), Institute of Atmospheric Physics, Chinese Academy of Sciences, Beijing, China

[2]Plateau Atmosphere and Environment Key Laboratory of Sichuan Province, School of Atmospheric Sciences, Chengdu University of Information Technology, Chengdu, China

[3]Center for Excellence in Regional Atmospheric Environment, Institute of Urban Environment, Chinese Academy of Sciences, Xiamen, China

[*]Corresponding author: Zirui Liu (liuzirui@mail.iap.ac.cn); Yuesi Wang (wys@mail.iap.ac.cn)

**ABSTRACT:** The simultaneous observation and analysis of atmospheric fine particles (PM$_{2.5}$) on a regional scale is an important approach to develop control strategies for haze pollution. In this study, samples of filtered PM$_{2.5}$ were collected simultaneously at three urban sites (Beijing, Tianjin, and Shijiazhuang) and at a regional background site (Xinglong) in the Beijing-Tianjin-Hebei (BTH) region from June 2014 to April 2015. The PM$_{2.5}$ at the four sites mainly comprised organic matter, secondary inorganic ions and mineral dust. Positive matrix factorization (PMF) demonstrated that, on an annual basis, secondary inorganic aerosol was the largest PM$_{2.5}$ source in this region, accounting for 29.2–40.5% of the PM$_{2.5}$ mass at the urban sites; the second largest PM$_{2.5}$ source was motor vehicle exhaust, particularly in Beijing (24.9%), whereas coal combustion was also a large source in Tianjin (12.4%) and Shijiazhuang (15.5%), and particularly dominated in winter. Secondary inorganic aerosol plays a vital role in the haze process, with the exception of the spring haze of Shijiazhuang and Tianjin, where the dust source was crucial. In addition to secondary transformations, local direct emissions (coal combustion and motor vehicle exhaust) significantly contribute to the winter haze at the urban sites. Moreover, with the aggravation of haze pollution, the OC/EC mass ratio of PM$_{2.5}$ decreased considerably and the nitrate-rich secondary aerosol increased during all four seasons in Beijing, both of which indicate that local motor vehicle emissions significantly contribute to the severe haze episodes of Beijing. To assess the impacts of regional transport on haze pollution, the PMF results were further processed with backward trajectory clusters analysis, revealing that haze pollution usually occurred when air masses originating from polluted industrial regions in the south prevailed and is characterized by high PM$_{2.5}$ loadings with considerable contributions from secondary aerosols. This study suggests that the control strategies to mitigate haze pollution in the BTH region should be focused on the reduction of gaseous precursor emissions from fossil fuel combustion (motor vehicle emissions in Beijing and coal combustion in Tianjin, Hebei and nearby provinces).

## 1 Introduction

Due to rapid economic development, rapid urbanization processes and excessive energy consumption, regional haze pollution has been recognized as the most severe environmental problem in China and has received extensive attention from the government, public and scientists in recent years (Zhang et al., 2015a). Haze pollution mainly occurs in economically developed urban agglomerations; the most seriously polluted regions are typically the Beijing-Tianjin-Hebei (BTH) region, the Yangtze River Delta (YRD) region, the Pearl River Delta region (PRD) and the Sichuan Basin (Zhang et al., 2012; Zhang and Cao, 2015). The BTH region, which includes the two megacities of Beijing and Tianjin as well as Hebei Province, has the highest density of coal consumption and heavily polluting industries in China and is surrounded by Shandong, Henan, Shanxi and Inner Mongolia, which are all heavily populated, industrialized, urbanized and are frequently reported to have serious haze pollution due to their intensive emissions of air pollutants (Liang et al., 2016; Wang et al., 2014a). Therefore, because the BTH region features the strongest pollutant emissions (Zhao et al., 2012), unfavorable meteorological conditions (Cai et al., 2017; Xu et al., 2011), and a unique topography, extreme haze pollution, characterized by high fine particulate matter ($PM_{2.5}$) loading and very low visibility, has frequently occurred in this region. From 2014-2015, of the 190 priority pollution monitoring cities in China, the annual average concentration of atmospheric $PM_{2.5}$ was highest in the BTH region (Zhang and Cao, 2015). Additionally, this region is characterized by frequent dust storms and corresponding high mineral dust loading episodes in spring (Huang et al., 2010; Sun et al., 2010).

Haze pollution with high fine PM loading could profoundly impact ecosystems, regional-scale atmospheric visibility, traffic safety, the economy, and interactions with climate (Zhang et al., 2015a); more importantly, this pollution can have adverse effects on human health, including the increased risks of respiratory, cardiac and other medical conditions (Elliot et al., 2016; Wu et al., 2017), thus leading to increased mortality rates, especially in megacities, which are generally seriously polluted and densely populated. In addition to the particle mass concentration and particle size, the health effects of PM are closely related to its chemical composition (Zhang et al., 2015a), and different diseases respond differently to different air pollutants (Tang et al., 2017). Moreover, the climate and environmental domino effects of PM are also closely related to the PM chemical compositions due to their different optical properties, such as those of black carbon, mineral particles, and brown carbon, which are light absorbing, while organic matter, ammonium sulfate and ammonium nitrate are light scattering (Tao et al., 2014a; Wang et al., 2015b; Wu et al., 2009; Zhang et al., 2016). These chemical constituents mainly originate from various anthropogenic sources, such as coal combustion, vehicle exhaust emissions, biomass burning, cooking, and industry-related emissions, among others. Therefore, the key to reducing $PM_{2.5}$ concentrations and improving air quality is to control these sources, which necessitates a strong demand for increased knowledge about the detailed chemical natures and

sources of $PM_{2.5}$ in the BTH region.

Haze pollution has significant regional characteristics. In addition to local emissions, the regional or inter-regional transport of primary PM and gaseous precursors plays an important role during haze periods (Chen et al., 2017; Li et al., 2017, 2015; Tao et al., 2012; Wang et al., 2014a; Ying et al., 2014). For example, the $SO_2$ measured in Beijing includes a large regional contribution transported from southern industrial areas (Guo et al., 2014). This contribution points to an urgent demand for wider collaborative works on emission control strategies between neighboring cities or provinces. For Shanghai, which is a megacity in the YRD region, Wu et al. (2017) estimated that the application of multiregional integrated control strategies in neighboring provinces could be most effective in reducing $PM_{2.5}$ in Shanghai and could largely reduce the economic losses caused by haze pollution. Extensive studies have been performed to investigate the formation mechanisms and emission sources of haze pollution in the BTH region and have obtained many valuable results (Du et al., 2014; Liu et al., 2016a; Sun et al., 2013; Wang et al., 2014b; Zhang et al., 2014; Zhao et al., 2013d). Massive anthropogenic emissions from diverse local sources, such as regional civil/industrial energy consumption, urban traffic, biomass burning and resuspended dust, and those transported from nearby provinces are widely regarded as the intrinsic reasons behind regional haze pollution events (Zhang et al., 2013; Zhao et al., 2012). Abnormal and unfavorable weather conditions also act as crucial factors in the formation of extensive and prolonged haze pollution events, such as the persistent haze event in January 2013 (Tao et al., 2014b; Wang et al., 2014b). In addition, many case studies, such as the winter regional haze events of 2010 (Zhao et al., 2013d) and 2013 (Sun et al., 2014; Wang et al., 2014b), have also revealed that severe haze events are largely driven by the high secondary production of sulfate, nitrate, ammonium and secondary organic aerosols (SOA), suggesting that aerosol chemistry plays a dominant role in haze evolution. Recent studies have reported a new efficient formation pathway for sulfate in the Beijing winter haze via reactive nitrogen chemistry in aerosol water during haze events (Cheng et al., 2016; He et al., 2014; Wang et al., 2016a, 2013b). In terms of haze mitigation strategies, Guo et al. (2014) suggested that regulatory controls of gaseous emissions for volatile organic compounds and nitrogen oxides from local transportation and sulfur dioxide from regional industrial sources are the keys to reducing the urban PM level in Beijing. However, these studies were often conducted at single sites (mostly in Beijing) and/or for short periods (specific haze events or a certain season); long-term multisite studies are scarce (Li et al., 2017; Shen et al., 2016; Zhang et al., 2013; Zhao et al., 2013c; Zong et al., 2016). In such studies, further questions are raised: first, due to the relatively few studies in Tianjin and Hebei, especially with respect to the source explorations of $PM_{2.5}$, we cannot fully understand the overall characteristics of the haze pollution in the BTH region; second, it is hard to directly compare the results between single-site studies to conduct a regional assessment as these studies covered different time periods and were conducted using different analytical approaches; third, the spatial and temporal variability of the

PM$_{2.5}$ sources in this region have not been extensively investigated, particularly with respect to the evolutions of emission sources at different pollution levels and their spatial variability. The above imperfections can limit the understanding of the sources and evolution processes of haze pollution on a regional scale and complicate effective mitigation strategies.

In this study, we conducted simultaneous measurements of PM$_{2.5}$ at three urban sites (Beijing, Tianjin, Shijiazhuang) in the BTH region and at one regional background site (Xinglong) as well as analyzing their chemical compositions and quantifying the apportionment of their sources using unified data processing and analytical methods. In addition, we emphatically analyzed the evolutions of the chemical compositions and emission sources at different pollution levels as well

as their seasonal and spatial differences. To further explore the influences of regional transport on the haze pollution, the source apportionment results in combination with the backward trajectory clustering was used. This study can provide an overall understanding of the regional signal of PM$_{2.5}$ pollution in the BTH region and support stakeholders and policy makers in understanding the impacts of regional sources on the high PM$_{2.5}$ loadings, thus facilitating the design of effective

joint emission abatement strategies.

## 2. Materials and methods
## 2.1 Sites and sampling

### 2.1.1 Site description

      Four sampling sites were selected in the Beijing-Tianjin-Hebei region (Fig. 1), including

three urban sites (Beijing, Tianjin and Shijiazhuang) and a regional background site (Xinglong). Beijing is the capital of China; Tianjin is an economically developed municipality; and Shijiazhuang is the capital of Hebei province, with an annual PM$_{2.5}$ concentration in 2013 ranking second in Hebei province and the whole of China (http://www.greenpeace.org.cn/PM25-ranking/). These three cities have their own atmospheric characteristics due to their different energy and

industrial structures, and thus, could represent the pollution characteristics of different types of urban areas in the BTH region. The Beijing site (39.97 °N, 116.38 °E) was situated in the courtyard of the Institute of Atmospheric Physics (IAP) of the Chinese Academy of Sciences (CAS). The Tianjin site (39.09 °N, 117.19 °E) was located in the Tianjin Atmospheric Boundary Layer Observatory of the Chinese Meteorological Administration, and the Shijiazhuang site

(38.03 °N, 114.53 °E) was located at the Hebei Meteorological Service. The sampling sites in Beijing, Tianjin and Shijiazhuang were affected by non-specific pollution sources while being influenced by mixed emission sources, such as local motor vehicle emissions, coal combustion, road dust, industrial activities, cooking, and transported pollutants, etc. Therefore, these sites are considered to be representative of typical urban environments. The sampling site in Xinglong

(40.39 °N, 117.58 °E) was located at Xinglong Observatory, National Astronomical Observatory, Chinese Academy of Sciences. Xinglong Observatory is located in the northeastern region of

Beijing with a liner distance of approximately 110 km from Beijing and is surrounded by mountains and thus allowing it to be minimally affected by human activities. Therefore, it is one of the regional atmospheric background stations of the Chinese Academy of Sciences.

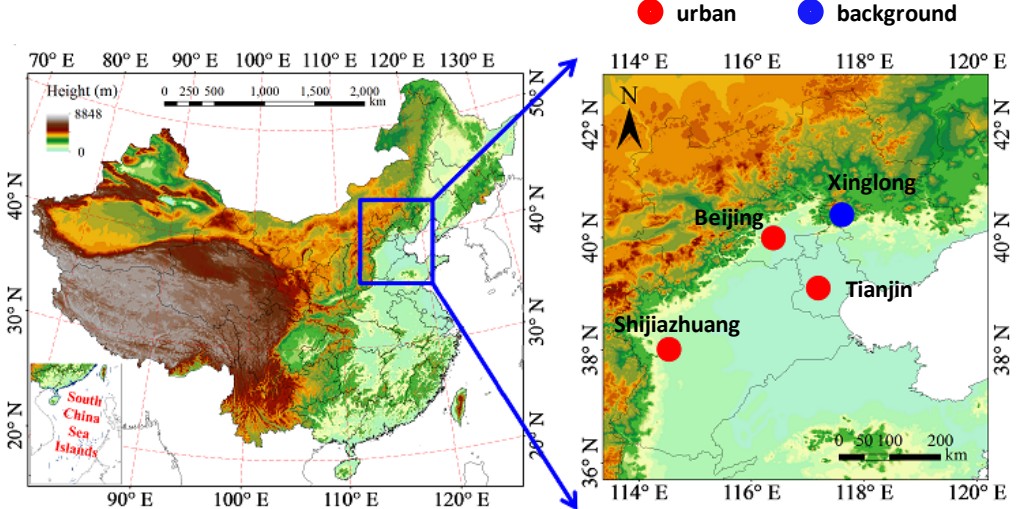

**Figure 1**. Map of the sampling sites (Beijing, Tianjin and Shijiazhuang are representatives of urban stations, whereas Xinglong represents the regional background)

## 2.1.2 PM$_{2.5}$ sampling

The PM$_{2.5}$ samples were simultaneously collected at the four sites using a PM$_{2.5}$ sampler (TH-150C, Tianhong, Wuhan) at an airflow rate of 100 L/min from June 2014 to April 2015. During each season, we collected PM$_{2.5}$ samples on 90 mm quartz membrane filters every day and night for one month, except on rainy days. The specific sampling period for the summer extended from 15 June 2014 to 14 July 2014, that of the autumn extended from 15 September 2014 to 14 October 2014, that of the winter extended from 29 December 2014 to 27 January 2015, and that of the spring extended from 20 March 2015 to 18 April 2015. The sampling time of each sample was 11.5 h, which generally occurred from 8:00 am to 19:30 pm during the daytime and from 20:00 pm to 7:30 am of the next day during the night. Over the entire observational period, 224, 214, 221 and 211 samples were collected in Beijing, Tianjin, Shijiazhuang and Xinglong, respectively, and numbers of samples in each season are shown in Table S1.

Detailed records of the instrumental conditions were collected during the sampling, including the sampling time, the sampled-air volume, atmospheric pressure, air temperature, etc. After sampling, the quartz filters were individually placed in petri dishes and were immediately stored at −20 °C prior to weighing and subsequent analysis. To ensure that the instrument worked at the specified flow rate, the airflow rate of the sampler was calibrated before and after each sampling. The carbon brush was replaced every month, and the outlet of the tail pipe was kept far away from the sampler in order to avoid contaminating the filter samples. During the sampling process, strict quality control was conducted to avoid any contamination. The frequent cleaning of the cutter and tray of the membrane was basic and necessary.

## 2.2 Filter analysis

### 2.2.1 Gravimetry

The quartz membrane filters, which were packaged with aluminum foil, were prefired in a muffle furnace at 500 ℃ for 4 h to remove organic material. In addition, in order to minimize the influence of water adsorption, the filters were weighed before and after sampling using a microelectronic balance with a reading precision of 10 μg after undergoing a 48 h equilibration period inside a chamber under the conditions of constant temperature (20±1 ℃) and humidity (45±5%). The atmospheric $PM_{2.5}$ masses were deduced from the gravimetric measurements performed before and after sampling. To guarantee the accuracy of the weights, the weighing was repeated until a difference of less than 0.10 mg between the two measured weights was achieved. All the procedures were strictly quality-controlled to avoid any possible contaminations of the samples.

### 2.2.2 Chemical analysis

A size of 2.011 $cm^2$ filter split from one quarter of each sample was ultrasonically extracted using 50 mL of deionized water (with a specific resistivity of 18.2 MΩ/cm) for 30 min. After passing through microporous membranes with pore sizes of 0.22 μm, the extracted solutions were analyzed using an ion chromatograph (IC) system (Dionex ICS-90, USA) for the detection of $SO_4^{2-}$, $NO_3^-$, $NH_4^+$, $Cl^-$, $K^+$, $Na^+$, $Ca^{2+}$ and $Mg^{2+}$. More details are given in Huang et al. (2016).

A 0.495 $cm^2$ punch from another quarter of each sample was used for the analysis of organic carbon (OC) and elemental carbon (EC), which was performed using a thermal/optical carbon aerosol analyzer (DRI Model 2001A, Desert Research Institute, USA) following the protocol of IMPROVE_A (TOR). In a pure helium atmosphere, OC1, OC2, OC3 and OC4 are produced stepwise at 140 ℃, 280 ℃, 480 ℃ and 580 ℃, respectively; followed by EC1 (540 ℃), EC2 (780 ℃) and EC3 (840 ℃) in a 2% oxygen-contained helium atmosphere. The OPC (organic pyrolyzed carbon) is determined when the reflected laser signal returns to its initial value after oxygen is added to the analyzed atmosphere. Therefore, the OC is operationally defined as OC=OC1+OC2+OC3+OC4+OPC, while the EC is defined as EC=EC1+EC2+EC3–OPC. Detailed procedures can be found in Xin et al. (2015).

The microwave acid digestion method was used to digest the filter samples into liquid solution for elemental analysis. One quarter of each filter sample was placed in the digestion vessel with a mixture of 6 mL $HNO_3$, 2 mL $H_2O_2$ and 0.6 mL HF, and was then exposed to a three-stage microwave digestion procedure from a microwave-accelerated reaction system (MARS, CEM Corporation, USA). After that, the digestion solution was transferred to PET bottles and diluted to 50 mL with deionized water (with a conductivity of 18.2 MΩ/cm). Agilent 7500a inductively coupled plasma mass spectrometry (ICP-MS, Agilent Technologies, Tokyo, Japan) was used to determine the concentrations of 18 trace elements in the digestion solution, including Mg, Al, K,

Ca, V, Cr, Mn, Fe, Co, Ni, Cu, Zn, As, Se, Ag, Cd, Tl and Pb. More detailed information, such as instrument optimization, calibration and quality control, is given in Wang et al. (2016b).

## 2.3 Data analysis method
### 2.3.1 Chemical mass closure

The chemically reconstructed $PM_{2.5}$ mass ($PM_{chem}$) was calculated as comprising eight categories of chemical species, which can be expressed as follows:

$$[PM_{chem}] = [Organic\ matter] + [EC] + [Mineral\ dust] + [Trace\ metals] + [Sulfate] + [Nitrate] + [Ammonium] + [Chloride]. \qquad (1)$$

In estimating organic matter (OM), an OC to OM conversion factor of 1.6 was adopted for the aerosols at urban sites (Cao et al., 2007; Turpin and Lim, 2001) and regional background site. Although the literature suggests that a higher OC to OM conversion factor of 2.1 is suitable for rural sites (Bressi et al., 2013; Turpin and Lim, 2001), we still used a uniform value of 1.6 for the sake of spatial comparisons. Therefore,

$$[OM] = 1.6 \times [OC]. \qquad (2)$$

The calculation of mineral dust was performed on the basis of crustal element oxides (such as $Al_2O_3$, $SiO_2$, $CaO$, $Fe_2O_3$, $TiO_2$, $MnO_2$ and $K_2O$) (Christoforou et al., 2000). The Ti content was very low in atmospheric particulate matter, with 0.04 μg/m$^3$ measured in the $PM_{2.5}$ in Beijing, Tianjin and Shijiazhuang (Zhao et al., 2013b). Thus, eliminating the Ti content has an almost negligible influence on the estimation of the mineral dust. Mineral dust was calculated as follows:

$$[Mineral\ dust] = [Al_2O_3] + [SiO_2] + [CaO] + [MnO_2] + [Fe_2O_3] + [K_2O] = 2.14 \times [Si] + 1.89 \times [Al] + 1.4 \times [Ca] + 1.58 \times [Mn] + 1.43 \times [Fe] + 1.21 \times [K] \qquad (3)$$

In this study, the measurements of the trace elements in the particles did not include the determination of Si, so the Si content was calculated based on its ratio to Al in crustal materials, namely, $[Si] = 3.41 \times [Al]$ (Mason, 1966). The calculations of $K_2O$ and $Fe_2O_3$ were also based on their ratios to Al in crustal materials (Wedepohl, 1995) since they have abundant artificial sources in addition to natural sources.

The trace metal content reflects the sum of 11 different heavy metal species and is expressed as:

$$[Trace\ metals] = V + Co + Ni + Cu + Pb + Zn + As + Se + Ag + Cd + Tl \qquad (4)$$

The above chemical reconstruction method was applied to the four sites, and comparisons of the reconstructed results ($PM_{chem}$) with the gravimetric results ($PM_{grav}$) are shown in the Supplement Fig. S1. It can be clearly seen that $PM_{chem}$ is significantly related to $PM_{grav}$, indicating that the chemical reconstruction method exhibited strong reliability. However, the $PM_{chem}$ concentrations at the four sites were all less than those of $PM_{grav}$; therefore, there exists unresolved matter that may largely be retaining water in the sampling membrane and particulate matter. Moreover, during the period between weighing and chemical measurements, the volatilization of

organic matter and the decomposition of ammonium nitrate may occur. The discrepancy between $PM_{chem}$ and $PM_{grav}$ was thus defined as Unknown.

### 2.3.2 Source apportionment

The EPA Positive Matrix Factorization (PMF) 5.0 model was applied to apportion the sources of $PM_{2.5}$ in this study, as it is an effective source apportionment receptor model that has been successfully applied for source apportionments in many cities and regions worldwide (Huang et al., 2014; Reff et al., 2007). Unlike to the CMB model (Chemical Mass Balance), the PMF model does not require source profiles prior to analysis, but only requires the values of the concentrations of the sample species and their uncertainties (U.S. Environmental Protection Agency, 2014; Zhang et al., 2015c). In this study, the model simulations were applied to datasets comprising 34 species: eight carbon fractions (OC1, OC2, OC3, OC4, OPC, EC1, EC2 and EC3), 8 inorganic species ($SO_4^{2-}$, $NO_3^-$, $NH_4^+$, $K^+$, $Na^+$, $Ca^{2+}$, $Mg^{2+}$ and $Cl^-$), and 18 trace elements (Mg, Al, K, Ca, V, Cr, Mn, Fe, Co, Ni, Cu, Zn, As, Se, Ag, Cd, Tl and Pb). Due to the low OC and EC concentrations at the background site, the entire concentrations of OC and EC were input into the PMF model instead of the eight carbon fractions. In addition to the concentrations of the sample chemical species, the uncertainties of the sample species were calculated based on two different situations according to the PMF 5.0 user guide (U.S. Environmental Protection Agency, 2014):

If the concentration is less than or equal to the provided method detection limit (MDL), the uncertainty is calculated using a fixed fraction of the MDL, which is written as $Uncertainty = \frac{5}{6}MDL$. If the concentration is greater than the provided MDL, the calculation is defined as $Uncertainty = \sqrt{(Error\ Fraction + Concentrtion)^2 + (0.05MDL)^2}$. In this study, the error fractions of $SO_4^{2-}$, $NO_3^-$ and $NH_4^+$ were estimated to be 5%, those of OPC, EC2 and EC3 were 15%, and those of other species were 10%. PMF analysis requires a complete data set, in order to reduce the error, the samples with missing values of individual species were excluded, rather than replaced by the mean concentrations of the remaining observations.

The number of factors must be chosen prior to using PMF. In this study, the PMF solutions using 5–12 factors at the three urban sites and 3–9 factors at the regional background site were explored with a final factor number chosen based on interpretability as well stability across bootstrap-replicate data sets (Xie et al., 2013a, 2013b).

### 2.4 Meteorological data and backward trajectory modeling

Meteorological data, including the ambient temperature, relative humidity and wind speed in Beijing and Xinglong were measured within 50 m of the filter sampling sites, using an automatic meteorological observation instrument (Milos520, Vaisala, Finland) located at an 8 m measurement height. In addition, in Tianjin and Shijiazhuang, the meteorological data was obtained from the meteorological monitoring stations of the China Meteorological Administration,

which was located within 100 m of the sampling sites.

The backward trajectory analysis method is widely applied to identify the potential source regions and transport pathways of air masses, especially for serious air pollution episodes (Gao et al., 2015; Hu et al., 2012; Zhang et al., 2014). In this study, 48 h backward trajectories terminated at a height of 100 m above ground level were calculated for the all four sampling sites using the Hybrid Single-Particle Lagrangian Integrated Trajectory (HYSPLIT 4.9) model developed by the U.S. National Oceanic and Atmospheric Administration/Air Resources Laboratory (NOAA/ARL). The trajectories were calculated every 12 h, with starting times at 8:00 and 20:00 local time (corresponding to each sampling) during the entire observational period.

## 3. Results and discussion

### 3.1 General characteristics of $PM_{2.5}$

#### 3.1.1 Annual mass concentrations

The temporal variability of the gravimetrically determined $PM_{2.5}$ concentrations at the four sites (Beijing, Tianjin, Shijiazhuang and Xinglong) throughout the entire observation period is shown in Fig. 2. The strong day-to-day variability of the $PM_{2.5}$ concentrations can be easily observed, especially in winter, when $PM_{2.5}$ concentrations range from 34.1 to 612.6 μg/m$^3$ in Shijiazhuang. These concentrations typically record periodic 'clean-polluted-clean' cycles for a few days, which were also reported by Guo et al. (2014), who noted that the Beijing haze pollution underwent clear periodic cycles of 4–7 days in length. These periodic cycles of haze episodes are primarily driven by atmospheric processes and fluctuations in meteorological conditions (Guo et al., 2014; Zhang et al., 2015a), such as wind speed, relative humidity, air temperature/pressure, atmospheric stability, the height of the planetary boundary layer and air mass origins. Very similar patterns of $PM_{2.5}$ temporal variations were found at all four sites (Fig. 2), suggesting homogeneous characteristics of atmospheric particulate matter on a regional scale.

On average, the annual $PM_{2.5}$ concentrations throughout the entire observation period recorded higher levels at the urban sites, which were 99.5±67.4, 105.7±63.1, 155.2±100.8 μg/m$^3$ in Beijing, Tianjin and Shijiazhuang, respectively, representing values that were 1.5, 1.6 and 2.4 times those at the background site (Xinglong), respectively. During the entire observational period, 81% of the samples in Shijiazhuang exceeded the second grade of the $PM_{2.5}$ daily average mass concentrations in China (75 μg/m$^3$), followed by Tianjin (63%) and Beijing (55%), with the minimum occurring in Xinglong (29%) (Table S1). Particularly serious pollution was observed in Shijiazhuang, which consumes huge amounts of energy for industrial processes and daily life (Zhao et al., 2013c) and exhibits higher relative humidity and lower wind speeds than the other sites (Table S2), which are both beneficial for the accumulation of $PM_{2.5}$ mass (Liu et al., 2017b). Moreover, the largest differences in the $PM_{2.5}$ average mass concentrations between the urban sites and the background site occurred in winter, yielding values of 2.2–4.1 times those in Xinglong.

This spatial difference can be explained by the strong intensity of pollution emissions (coal combustion for heating) in winter at the urban sites.

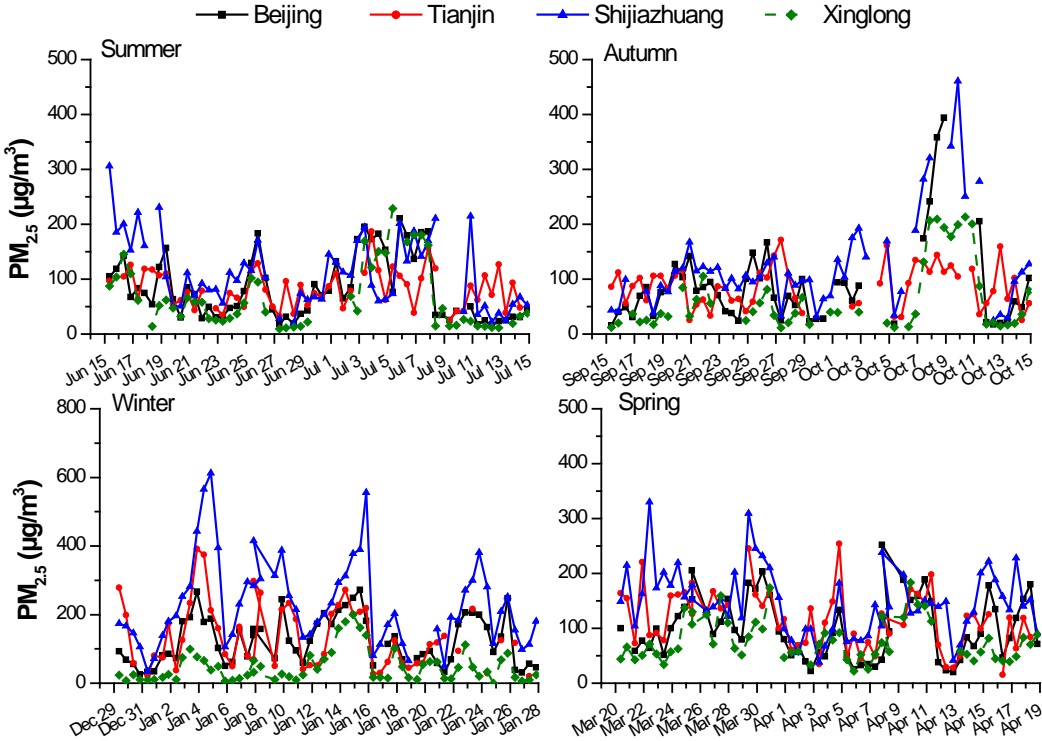

**Figure 2**. Time-series of the gravimetric PM$_{2.5}$ concentrations during the four study periods

### 3.1.2 Seasonal variation

Due to the minor effects of anthropogenic emissions, the seasonal variations of PM$_{2.5}$ concentrations in Xinglong were not significant (56.7–77.9 μg/m$^3$), and only slightly higher values were observed in spring. Zhao et al. (2009) also determined that the maximum PM$_{2.5}$ concentrations usually occur in spring in Shangdianzi, which is another regional background area in the North China Plain. However, at the urban sites, significant seasonal variations were observed, especially in Shijiazhuang. The highest PM$_{2.5}$ values were recorded in winter, with average concentrations of 124.8±69.9, 136.6±93.8 and 231.8±129.1 μg/m$^3$ in Beijing, Tianjin and Shijiazhuang, respectively. There were four extreme haze episodes that occurred in Shijiazhuang, particularly serious were the episodes from January 2 to 6 and from January 11 to 16. Numerous studies have also revealed that the heaviest haze pollution events with extremely high PM$_{2.5}$ loadings occurred in winter in the BTH region (Wang et al., 2012a; Zhang and Cao, 2015; Zhao et al., 2013d), which has mainly been attributed to the combination of intensive coal-fired heating and unfavorable meteorological conditions (i.e., more frequent occurrences of stagnant weather, temperature inversions and low boundary layer heights) in this region (Tang et al., 2016a; Zhang and Cao, 2015; Zhao et al., 2009). Following winter, the average PM$_{2.5}$ concentrations in spring also remained at a relative high level at the urban sites (101.0–148.4 μg/m$^3$), which may have

partly resulted from the enhanced mineral dust (see Fig. 4) produced by the relatively high wind speeds during this season (Table S2). Due to strong turbulence that occurs under conditions of strong radiation intensity and high temperature in summer, as well as the high atmospheric mixing layer generally observed in this season (Tang et al., 2016a), air pollutants could have been effectively diluted and diffused, thus resulting in the lowest $PM_{2.5}$ concentrations being measured during this season. The $PM_{2.5}$ concentrations in autumn were close to those in summer, but one extreme haze episode was recorded from October 5 to 11 in Beijing and Shijiazhuang, with the highest concentration of the daily $PM_{2.5}$ reaching 394 μg/m$^3$ in Beijing and 460 μg/m$^3$ in Shijiazhuang. The extreme haze episode in October 2014 was recorded and analyzed in depth by Yang et al. (2015).

## 3.2 Chemical compositions of $PM_{2.5}$
### 3.2.1 Annual compositions

The chemical compositions of the entire sample set collected from the three urban sites and the regional background site were similar, further confirming the regional homogeneity of atmospheric $PM_{2.5}$. The $PM_{2.5}$ in this region (Fig. 3) primarily comprised organic matter (OM=OC×1.6, 16.0–25.0%), secondary inorganic ions (SNA, including sulfate, nitrate and ammonium, 43.6–53.3%), mineral dust (14.7–20.8%), and lower proportions of EC (2.8–6.2%), chloride (1.9–5.5%) and trace metals (0.4–0.6%). The annual average concentrations of carbonaceous aerosols (OM plus EC) were 31.1, 27.0 and 44.2 μg/m$^3$ in Beijing, Tianjin and Shijiazhuang, respectively, thus constituting large fractions (25.5–31.2%) of $PM_{2.5}$ in the urban atmosphere. It is worth noting that EC accounted for 6.2%, 5.9% and 5.7% of the measured masses of $PM_{2.5}$ in Beijing, Tianjin and Shijiazhuang, respectively, which were higher than that measured in Xinglong (2.8%), reflecting the strong emissions from fossil fuel combustion in urban areas. In Xinglong, lower concentrations and lower fractions of OM but higher fractions of mineral dust and SNA than those of the urban sites were discovered. The presence of lower contributions of OM but higher contributions of SNA at the background site is consistent with the results measured on Qimu Island (another regional background site in northern China, which is located approximately 300 km southeast of the BTH region) (Zong et al., 2016). This was mainly attributed to the regional-scale emission characteristics of gaseous precursors in this region, in which there are more abundant $SO_2$, $NO_x$, $NH_3$ emissions than OC emissions (Zhao et al., 2012), and the general characteristics of the regional atmosphere are well reflected at the regional background site.

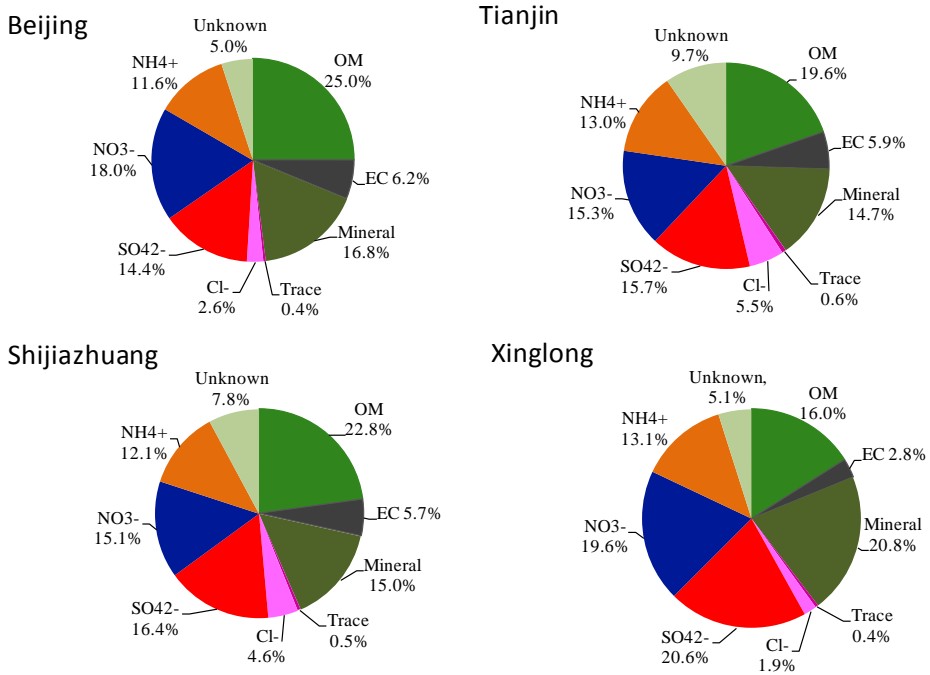

**Figure 3**. Pie charts depicting the percentages of the major chemical components in

gravimetric PM$_{2.5}$ based on annual data

As the key PM$_{2.5}$ constituents, sulfate, nitrate and ammonium (SNA), are generally
recognized to originate from the secondary conversions of gaseous SO$_2$, NO$_x$ and NH$_3$ via
gas-phase chemical reactions and heterogeneous reactions (Wang et al., 2016a; Zhang et al.,
2015a). In this study, they accounted for 14.4–20.6%, 15.1–19.6% and 11.6–13.1% of the annual
average PM$_{2.5}$ concentrations, respectively. Among the three urban sites, the highest NO$_3^-$
contribution (18.0%) was found in Beijing, which agrees well with it having a strong traffic source.
In addition, from the annual results, the mass ratio of NO$_3^-$/SO$_4^{2-}$ in Beijing was 1.25, which was
higher than that in Tianjin (0.97) and Shijiazhuang (0.92) as well as that in Xinglong (0.95). The
NO$_3^-$/SO$_4^{2-}$ mass ratio has often been used as an indicator of the relative contributions of sulfur
and nitrogen from mobile versus stationary sources to aerosol particles in the atmosphere
(Arimoto et al., 1996; Cao et al., 2009; Han et al., 2016), as vehicle exhausts and coal-combustion
emissions are significant contributors of nitrate and sulfate, respectively (Huang et al., 2014).
Therefore, the higher NO$_3^-$/SO$_4^{2-}$ mass ratio of Beijing implies that the predominance of motor
vehicle emissions in the contributions to PM pollution (Han et al., 2016; Yang et al., 2015), while
in Tianjin and Shijiazhuang, coal combustion may still play a dominant role. However, compared
to the reported results, the NO$_3^-$/SO$_4^{2-}$ mass ratios of the two cities also increased from 0.85
(2009–2010) (Zhao et al., 2013b) to 0.92 (this study) in Shijiazhuang, as well as from 0.69 (2008)
(Gu et al., 2011) to 0.75 (2009–2010) (Zhao et al., 2013b) and to 0.97 (this study) in Tianjin. This
increasing trend also occurred in the background area, as the NO$_3^-$/SO$_4^{2-}$ mass ratio in Xinglong
increased from 0.78 (2009–2010) (Li, et al., 2013) to 0.95 (this study). These results clearly

revealed that atmospheric nitrate pollution is worsening in this region, which is generally recognized as being caused by increasing motor vehicle emissions and indicates the remarkable effect of the control measures of $SO_2$ emissions.

In addition to carbonaceous and SNA aerosols, mineral dust was also a major component of $PM_{2.5}$, constituting a smaller fraction of the $PM_{2.5}$ masses (14.7–16.8%) at the urban sites than at the background site (20.8%). $Cl^-$ exhibited higher concentrations and fractions in Shijiazhuang (7.2 μg/m$^3$, 4.6%) and Tianjin (5.8 μg/m$^3$, 5.5%) than it did in Beijing (2.6 μg/m$^3$, 2.6%) and Xinglong (1.2 μg/m$^3$, 1.9%), further illustrating the important contribution of coal combustion emissions to the $PM_{2.5}$ concentrations in Shijiazhuang and Tianjin. The concentration of trace metals varied from 0.3 to 0.7 μg/m$^3$ and constituted only a minor fraction of the masses.

### 3.2.2 Seasonal variations

The PM mass and its chemical compositions are governed by chemical processes, evolutions of emission sources and meteorological conditions (Bressi et al., 2013; Liu et al., 2017b), which usually exhibit seasonality. The seasonal patterns of $PM_{2.5}$ at the urban sites were mainly driven by OM, SNA and mineral dust, which were the major components of $PM_{2.5}$ during each season. In winter, the dominant component at the urban sites was OM; moreover, Figure 4 shows that the OM concentration and its contribution to $PM_{2.5}$ mass were significantly higher in winter (38.1–82.7 μg/m$^3$, 27.9–35.7%) than those in other seasons. In addition, the EC concentration also reached a maximum value in winter at the urban sites, yielding values of 11.8, 10.7 and 16.3 μg/m$^3$ in Beijing, Tianjin, and Shijiazhuang, respectively. Similar seasonal variations of carbonaceous aerosols were also observed in the BTH region (Beijing, Tianjin, Shijiazhuang, Chengde and Shangdianzi) (Zhao et al., 2013), Jinan (Yang et al., 2012) and the Pearl River Delta region (Cao et al., 2004). There are two possible explanations for this phenomenon. On the one hand, a substantial increase in the amount of coal-fires used for residential heating in winter could increase the abundance of carbon-containing emissions, including primary organic carbon, EC, and VOCs (Zhao et al., 2012). On the other hand, the lower temperatures in winter could favor the conversion from gaseous VOCs to their particulate forms (Wang et al., 2015), whereas the high temperatures in warm seasons, especially those in summer (during which the lowest OM concentrations can be seen in Fig. 4), may cause the semi-volatile organic compounds to mainly exist in their gaseous forms in the atmosphere. $Cl^-$, which is a good tracer of coal combustion, also exhibited higher concentrations and contributions to the $PM_{2.5}$ mass in winter (5.3–14.6 μg/m$^3$, 4.6–6.3%) than it did in other seasons (1.0–5.6 μg/m$^3$, 1.2–4.9%) at the urban sites. However, since it is less affected by local anthropogenic sources, the Xinglong site recorded the lowest concentrations and contributions of EC and $Cl^-$, which showed no distinct seasonal variations.

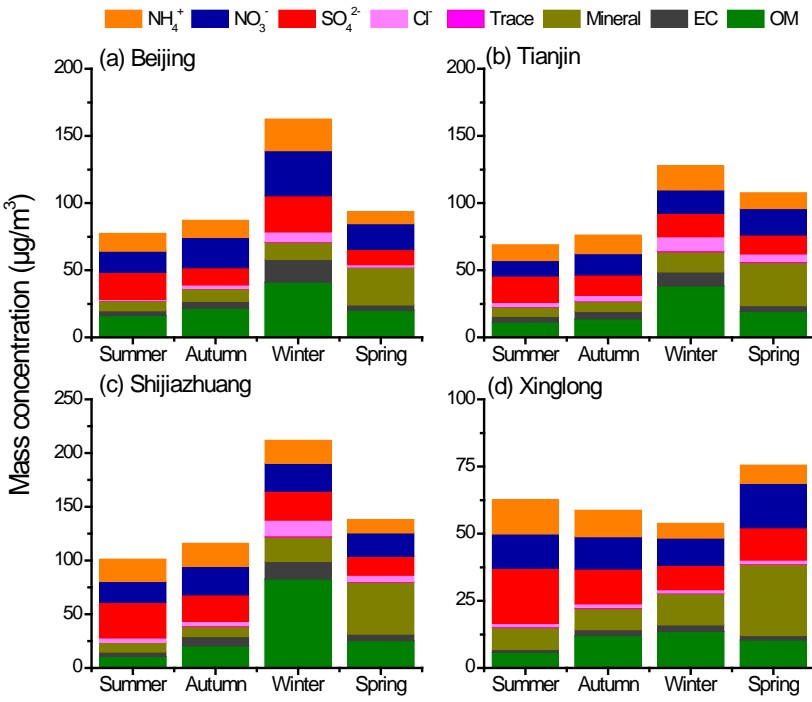

**Figure 4**. Seasonal variations of the major chemical components of $PM_{2.5}$ at the four sites

Unlike OM, SNA has its highest contributions in summer at the urban sites (51.7–66.2%), which were significantly higher than those in winter and spring. The prominence of SNA in summer was more apparent in Xinglong (71.2%), which reflects the dominant contributions of meteorological factors. At the urban sites, SNA also presented prominent contributions to the $PM_{2.5}$ in autumn (51.9–58.1%), and the average $PM_{2.5}$ concentrations were comparable in summer and autumn, which may have resulted from the high relative humidity in autumn, which is even higher than that in summer (Table S2). However, the contributions of sulfate and nitrate exhibited obvious seasonal differences at these three urban sites and even more apparent differences at the background site, recording greater contributions of sulfate in summer (23.6–29.9%) and nitrate in autumn (18.4–25.7%). This pattern was also found in our previous study in Beijing (Huang et al., 2016). This trend is closely related to their respective chemical/physical properties and mechanisms of generation, as nitrate tends to be decomposed under high temperatures (which mainly occurs in summer) due to the thermodynamic instability of ammonium nitrate, while the process of the chemical generation of ammonium sulfate (i.e., the gas-phase oxidation of $SO_2$ and its subsequent heterogeneous reactions) is largely promoted under the high temperatures and intense solar radiation of summer (Huang et al., 2016; Ianniello et al., 2011; Zhang et al., 2015a).

In spring, the primary chemical component at all four sites was mineral dust, which contributed 27.5–34.1% to $PM_{2.5}$ and was significantly higher than it was in other seasons (7.5–20.7%), thus reflecting the important influence of northwest dust transport on the atmospheric fine particles of the BTH region in that spring and the increase of local resuspended dust (such as road dust and construction dust) resulting from the enhanced wind speeds during this

season (Table S2).

## 3.3 Source apportionment using PMF

The sources/factors of PM$_{2.5}$ were apportioned by applying the PMF receptor model at three urban sites and at the background site for comparison. The identification of the sources was based on certain chemical tracers that are generally presumed to be emitted by specific sources and are present in significant amounts in the collected samples (Singh et al., 2016). Based on this, eight factors were identified for Beijing and Tianjin, nine were identified for Shijiazhuang, and only five were identified for Xinglong. The relative dominances of each source varied by site and season. Contributions of the identified sources determined by analyzing the annual data are shown in Fig. 6; the factor profiles of PM$_{2.5}$ for the regional background site (Xinglong) are listed in Fig. 5, while those for the urban sites are shown in Fig. S2–4. These factors can be summarized as follows:

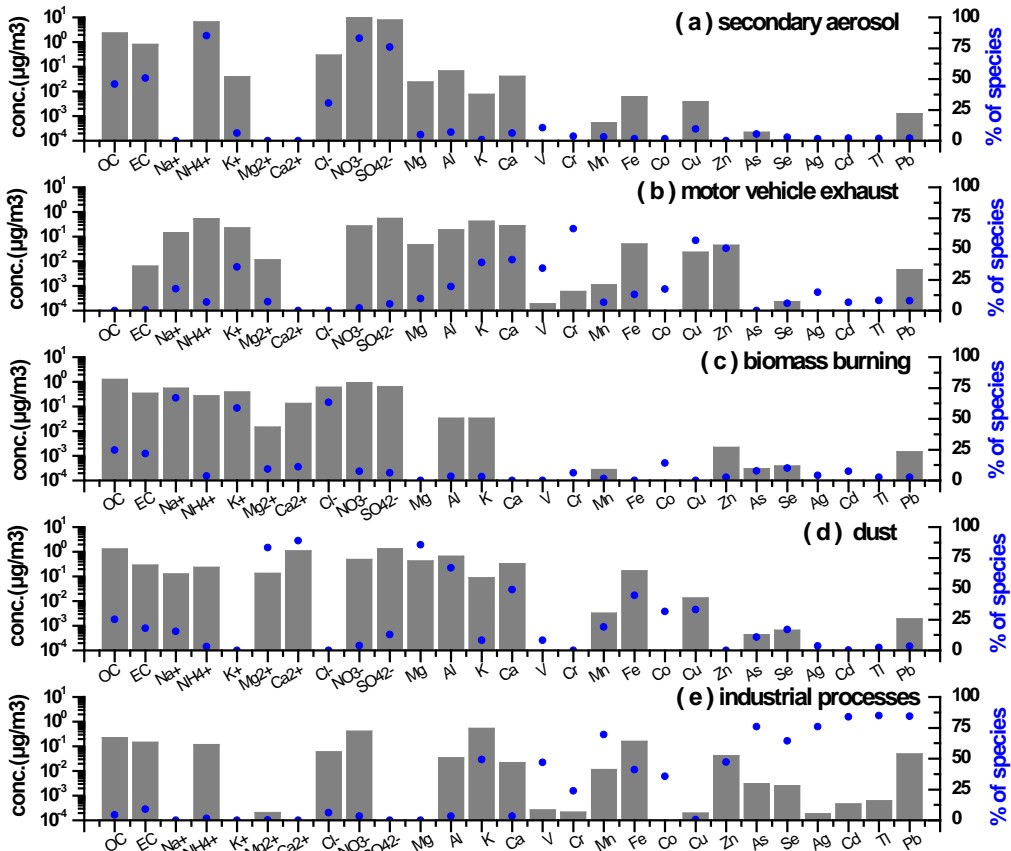

**Figure 5**. PMF factor/source profiles for the PM$_{2.5}$ samples throughout the entire study period in Xinglong in terms of concentrations (μg/m$^3$) and percentages (%)

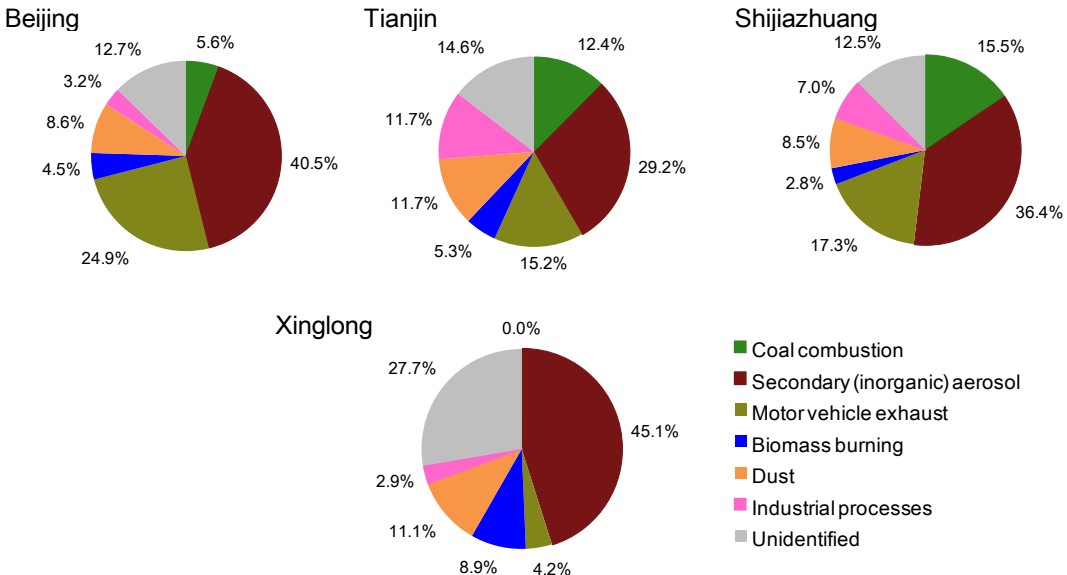

**Figure 6**. The annual contributions of the identified sources to PM$_{2.5}$ masses at the four sites

(i) Coal combustion. In China, coal combustion is generally used in thermal power plants, for industrial fuel use, as well as for winter residential heating in its northern cold regions. This source is characterized by high loadings of OC, EC, and chloride, most of which were apportioned in this factor (Fig. S2–4a). At the three urban sites, coal combustion source exhibited significantly higher contributions to the PM$_{2.5}$ (14.1%, 23.3% and 29.2% in Beijing, Tianjin and Shijiazhuang, respectively) in winter than were seen in other seasons (0.7–10.7%). This was strongly aligned with the seasonal characteristics of coal combustion activities in this region. The annual average emissions from coal combustion contributed only 5.6% to the PM$_{2.5}$ in Beijing and were much higher in Tianjin (12.4%) and Shijiazhuang (15.5%) but were not identified at Xinglong.

(ii) Secondary aerosol/inorganic aerosol. The dominant source was secondary inorganic aerosol at the three urban sites (29.2–40.5%) and secondary aerosol in Xinglong (45.1%). In Xinglong, this factor (Fig. 5a) can be identified as secondary aerosol because of the high contributions and accumulations of OC, sulfate, nitrate and ammonium, which caused the aerosol to include secondary inorganic aerosol (SIA) and secondary organic aerosol (SOA). In addition, approximately 30% of the total chloride was assigned to this source, indicating that coal combustion source was included in the secondary aerosol source. However, our analysis indicates that there were only very minor local coal combustion emissions in Xinglong. Therefore, this contribution can probably be attributed to the regional transport of coal combustion emissions, along with secondary source.

In Tianjin, high contributions of sulfate, nitrate and ammonium, which comprise most of the aerosol mass concentrations, were apportioned in the secondary aerosol (Fig. S3b) with only minor OC masses; thus, we identified this factor as secondary inorganic aerosol. In contrast, in Beijing and Shijiazhuang, secondary inorganic aerosol was further separated into two sources,

which are defined as nitrate-rich secondary (Fig. S2b and S4b) and sulfate-rich secondary aerosol (Fig. S2c and S4c), which record the respective characteristics of the prominent contributions of ammonium/nitrate and ammonium/sulfate. Consistent with the generating mechanisms and seasonal characteristics of nitrate and sulfate, the contribution of nitrate-rich secondary aerosol to

$PM_{2.5}$ had the highest values in autumn, whereas the sulfate-rich secondary aerosol had the highest contribution values in summer in Beijing and Shijiazhuang.

(iii) Motor vehicle exhaust. Emissions from motor vehicles are major factors in serious air pollution, especially in economically developed megacities. In our study, the factor of motor vehicle exhaust, which has high concentrations of OC, EC, and the trace metals of Cu, Zn and Pb,

which are considered to be characteristic species of bake wear dust and tire wear dust (Gao et al., 2014; Karnae and John, 2011; Tian et al., 2016; Zhang et al., 2013), was identified at all four study sites, contributing 24.9% (Beijing), 15.2% (Tianjin) and 17.3% (Shijiazhuang) of the aerosols at the urban sites and only 4.2% of those in Xinglong. This suggests the important role of motor vehicle exhaust in the urban $PM_{2.5}$ pollutions, especially in Beijing, where the number of motor

vehicles increased to 5.7 million by 2016. Notably, secondary aerosols are mainly produced by the gas-to-particle transformations of $SO_2$, $NO_x$, $NH_3$, and VOCs, and motor vehicle exhaust is an important source of the emissions of $NO_x$  and VOCs in urban areas (Huang et al., 2011; Tang et al., 2016b). Therefore, the actual contribution of motor vehicle exhaust to aerosols should be higher when considering the secondary formations of gaseous exhaust in the atmosphere.

Since 2017, the vehicular emission standard of China in Phase V (equivalent to European V) has been implemented on a national scale, and has caught up with developed countries. However, the less restrictive standards for oil quality in China than those in Europe and United States are the main reason for the strong motor vehicle emissions, particularly the limit standards of aromatics and alkenes. These two unsaturated hydrocarbon species have important effects on air quality

(Schell et al., 2001), as decreasing alkenes content can decrease the fire temperatures and reduce $NO_x$ emissions (Tang et al., 2015b). A new study has also reported that gasoline aromatic hydrocarbons had an essential role in urban SOA production enhancement and thus, significantly affected the ambient $PM_{2.5}$ (Peng et al., 2017). The limit standards of aromatic and alkene contents in the vehicle gasoline are respectively 40% and 28% for the China IV, implemented starting in

2014; 40% and 24% for the China V, starting in 2017; 35% and 18% for China VI, which is suggested for implementation in 2019 (http://www.nea.gov.cn/). In contrast, the limit standards of 35% for aromatics and 18% for alkenes (European IV) in Europe were implemented in 2005. In addition to the lower standards for oil quality, the phenomenon of substandard oil products also exists, as Tang et al. (2015b) reported that 48.4% of the gasoline samples in northern China

exceeded the aromatics limit standard (40%).

(iv) Biomass burning. Biomass burning emissions in northern China are mainly produced by the burning of agricultural straw and thus, often appear during the farming and harvest seasons.

These emissions can have a significant impact on the atmospheric chemistry and the climate on both a regional and global scale (Duan et al., 2004; Li et al., 2010; Sun et al., 2016). The source profiles of the factors defined as biomass burning (Fig. 5c, Fig. S2e, S3d and S4e) were aerosols rich in $K^+$, which is widely regarded to be a good tracer of biomass burning sources. In addition to $K^+$, the fresh smoke plumes of burning biomass also contain significant amounts of $Na^+$, $Cl^-$, OC and EC (Wang et al., 2013a), which were also found in the profiles of biomass burning in this study. The annual average contribution of biomass burning to $PM_{2.5}$ showed higher values in Xinglong (8.9%) than it did at the three urban sites (2.8–5.3%). In addition to its high proportion during the harvest season (autumn, 11.6%), biomass burning emissions exhibited their highest contributions to $PM_{2.5}$ in winter in Xinglong (14.6%) and recorded low values (1.0–4.4%) in winter at the three urban sites. This pattern can likely be attributed to the fact that a single type of fuel is used by the surrounding rural residents, as bio-fuels (i.e., straw and dry wood) are always utilized for cooking and winter heating (Zhao et al., 2012), which is totally different than the matter used for energy (mainly coal and natural gas) by urban and suburban residents. Similar to the motor vehicle source, the contribution of biomass burning would be higher if the emissions of secondary aerosol precursors (VOCs, $SO_2$ and $NO_x$), especially VOCs, are considered (Bo et al., 2008; Li et al., 2014; Yuan et al., 2010).

(v) Dust. Road dust from local traffic and construction activities (with abundant concentrations of $Mg^{2+}$, $Ca^{2+}$ and motor vehicle-related species such as Cu, Zn and Pb) (Han et al., 2007) and soil dust, which is mainly derived from long-range transport (and is more enriched in Al, Ca, Fe, Mg and Mn), were summarized as dust. This source was found to have an obvious seasonality, exhibiting its highest contributions in spring at the urban (17.2–21.0%) and background sites (22.2%). Influenced by the dust from the northwest, this seasonal variation was most significant and regular for soil dust. The factor of road dust was identified at three urban sites, which contributed annual average values of 3.5–7.8% of the $PM_{2.5}$ but was not extracted from the dust source in Xinglong due to the minor influence of anthropogenic sources.

(vi) Industrial processes. A striking feature of this source was its relatively high concentrations of the mining-related elements, such as V, Mn and Fe, and elements related to pollution produced by industrial processes, including As, Se, Ag, Cd, Tl and Pb. More than 50% of the mass concentrations of the above pollution elements were allocated to this source. The annual average emissions from industrial processes contributed 3.2–11.7% to $PM_{2.5}$ at the urban sites, which was lowest in Beijing. The industrial processes source in Xinglong (2.9%) may have been the result of the regional transport from regional cities with heavy industrial activity. In Tianjin and Shijiazhuang (two heavily industrial cities), an oil refining/metal smelting source, characterized by high concentrations of V, Mn, Fe, Co and Ni (Mohiuddin et al., 2014), was extracted from the emission source of industrial processes, contributing 2.8% (Tianjin) and 0.7% (Shijiazhuang) to the $PM_{2.5}$.

In summary, secondary inorganic aerosol (40.5%) and motor vehicle exhaust (24.9%) were the largest $PM_{2.5}$ sources in Beijing and have greater values than the results of the 2010 study by Wu et al. (2014). The contribution of motor vehicle exhaust was close to that provided by the Beijing Municipal Environmental Protection Bureau (19.9–22.4%, http://www.bjepb.gov.cn/) (Table S3). Compared with those of Beijing, there were more complicated sources of $PM_{2.5}$ in Tianjin and Shijiazhuang. Motor vehicle exhaust was also an important source at the two sites, but the contribution (15.2% in Tianjin and 17.3% in Shijiazhuang)was lower than that in Beijing, which was consistent with the results published by the Environmental Protection Bureau (EPB) (Table S3). However, coal combustion became an important source in Tianjin (12.4%) and Shijiazhuang (15.5%), which was slightly lower than that from the Environmental Protection Bureau and that given by Liu et al. (2017a) (16.5% in Tianjin). The biggest $PM_{2.5}$ source in Tianjin (29.2%) and Shijiazhuang (36.4%) was still secondary inorganic aerosol, of which the contribution in Tianjin was close to that of Liu et al. (2017a) (26.1%).

## 3.4 Evolution at different pollution levels

By using the $PM_{2.5}$ pollution grading standards of the Air Quality Index (AQI) technical regulations (HJ 633-2012) formulated by the Chinese Ministry of Environmental Protection as a reference, and considering the quantity of samples analyzed in this study, days with average concentrations of $PM_{2.5} < 75$, $75 \leq PM_{2.5} < 150$ and $PM_{2.5} \geq 150$ μg/m$^3$ were defined as clean, moderate pollution and heavy pollution days, respectively. The seasonal distributions of sample quantities at different pollution levels are listed in Table S1.

### 3.4.1 Evolution of chemical components

The evolutions of the chemical compositions at different pollution levels during the entire observational period and in each season are shown respectively in Fig. 7 and Fig. S5, from both of which we can see that nearly all of the chemical components increased continuously and noticeably with the aggravation of pollution. A remarkable increase of carbonaceous aerosols was observed during the pollution process, in which the annual average OC and EC concentrations on heavy pollution days were 3.5–7.0 and 4.6–5.9 times as high as they were on clean days, respectively. On an annual average, the OC/EC mass ratio decreased significantly with the pollution levels in Beijing, with its value varying from 3.4 (clean days) to 2.9 (moderate pollution days) to 2.1 (heavy pollution days). Tianjin also recorded a similar but milder pattern (varying from 2.3 to 2.0 to 1.9, respectively). This trend at the two sites has also been found in the investigation of specific pollution processes in summer from Jun 29 to Jul 8, in winter from Jan 11 to 16, and in spring from Mar 23 to Apr 1 (The extreme haze episode in autumn from Oct 5 to 11 was not analyzed due to many missing samples) (Fig. 8). As reported by Watson et al. (2001), lower OC/EC ratios are emitted from motor vehicles (1.1) than are emitted from coal combustion

(2.7) and biomass burning (9.0). Saarikoski et al. (2008) have also documented an OC/EC ratio of 6.6 for biomass burning and of 0.71 for traffic emissions. Therefore, we speculate that this pattern of variations of the OC/EC ratio in Beijing may be influenced by the strengthened contributions of local motor vehicle exhaust under heavily polluted conditions due to weakened regional transports, which usually contribute most during the initial and growth stages of haze episodes, while decreasing during the peak pollution stage. This mechanism has been confirmed in some specific pollution processes in Beijing (Liu et al., 2016b; Tang et al., 2015a). By using CAMx (Comprehensive Air Quality Model with Extensions), Wang et al. (2017) also documented that the extreme haze episode during January 2013 in urban Beijing was dominated by local contributions. Therefore, the fact that the OC/EC ratio decreases with the increasing development of haze pollution indicates the key role of local traffic emissions in the haze processes of Beijing.

In addition, SNA generally significantly contribute to the enhancement of the $PM_{2.5}$ mass during pollution events, particularly in Beijing and Xinglong as the contribution of SNA increased remarkably from 34.5% on clean days to 44.2% on moderate pollution days to 51.3% on heavy pollution days in Beijing and from 41.2% to 57.5% to 68.3%, respectively, in Xinglong (Fig. 7). Sulfate, nitrate and ammonium all clearly increased with the aggravation of pollution in Xinglong, suggesting that its high $PM_{2.5}$ loadings mainly resulted from the intensifying secondary transformations of gaseous pollutants ($SO_2$, $NO_x$ and $NH_3$) during stagnant meteorological conditions in the background area. However, in Beijing, only the contribution of nitrate (except in summer), the formation of which is local dominant (Guo et al., 2010), recorded a pronounced increase during the pollution process (Fig. S5), thus indicating that the haze pollution in Beijing mainly resulted from the secondary transformation of $NO_x$ that was mainly derived from local traffic emissions, once again reflecting the dominant contribution of local motor vehicle exhaust to Beijing haze episodes. An average increase in the contributions of SNA during the pollution process was not observed but did occur in all seasons except for spring in Tianjin and Shijiazhuang (Fig. S5), while the mineral dust contribution increased considerably in spring, indicating the important role of dust in the formation of spring haze at the two study sites.

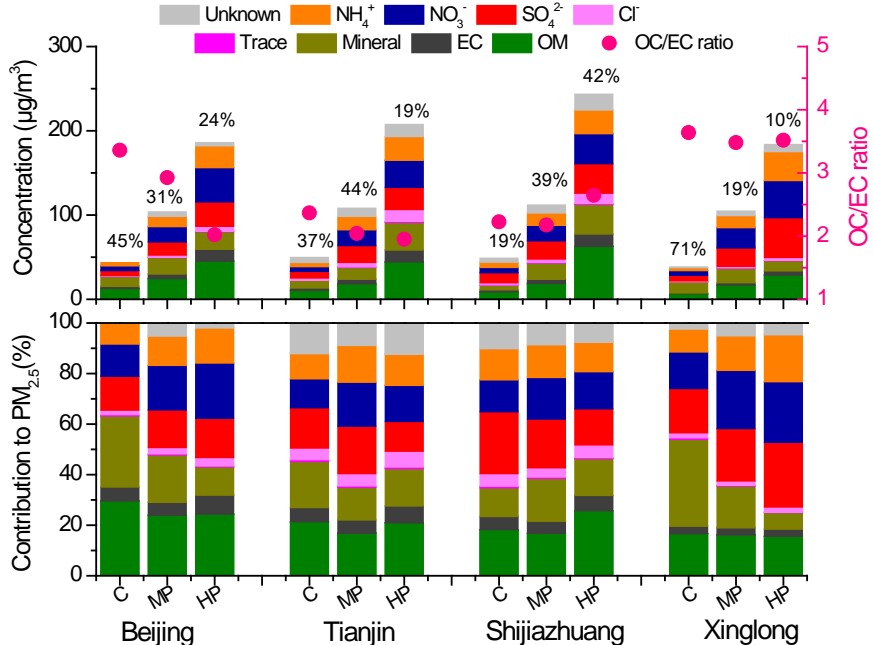

**Figure 7**. The evolutions of aerosol chemical species and the OC/EC mass ratio (marked with pink dots) at different pollution levels during the entire observational period. C, MP, and HP represent clean days ($PM_{2.5} < 75$ μg/m$^3$), moderate pollution days ($75 \leq PM_{2.5} < 150$ μg/m$^3$) and heavy pollution days ($PM_{2.5} \geq 150$ μg/m$^3$), respectively. "%" represents the proportion of the filter sample quantity at each pollution level out of the total samples.

From the above analysis, it can be seen that the chemical characteristics of haze pollution varied by site and season. The most prominent feature in summer was the intensive formation of SNA, which was observed simultaneously at the three urban sites on a regional scale during the period from Jun 29 to Jul 8 (Fig. 8a). The SNA contributed to the elevation of $PM_{2.5}$ concentration; SNA showed a substantial increase of a factor of 5.7, from 22.6 μg/m$^3$ during the clean period to 128.7 μg/m$^3$ during the haze episodes (Jul 2-N to Jul 4-N and Jul 5-N to Jul 7-N) in Beijing, and a corresponding increase of the contribution of SNA to $PM_{2.5}$ from 45.7% to 72.5%; the enhancement ratios were 4.4 and 3.4 in Tianjin and Shijiazhuang, respectively. The large increase in the concentrations and contributions of SNA during the haze process was observed in many previous studies and is mainly attributed to the enhanced secondary conversions via the enhanced heterogeneous reactions under relatively high humidity conditions during the haze periods (Fig. S6, averaging 67% during haze episodes and 37% during clean periods in summer in Beijing, as measured in this study) (Huang et al., 2016; Sun et al., 2013; Wang et al., 2012b; Yang et al., 2015). The degree of secondary formation of sulfate and nitrate is commonly estimated using the sulfur oxidation ratio (SOR=n-$SO_4^{2-}$/n-$SO_4^{2-}$+n-$SO_2$, where n refers to molar concentration) and the nitrogen oxidation ratio (NOR=n-$NO_3^-$/n-$NO_3^-$+n-$NO_2$), respectively (Huang et al., 2016; Zhao et al., 2013d). Higher values of the SOR and NOR indicate that more gaseous $SO_2$ and $NO_2$ would be oxidized to sulfate and nitrate in the atmosphere. In this study, the SOR and NOR were

significantly elevated during the increases of $PM_{2.5}$ concentrations during the summer period and remained at a high level during haze episodes in Beijing (averaging 0.92 for SOR and 0.38 for NOR), Tianjin (0.83 and 0.41) and Shijiazhuang (0.65 and 0.44). In autumn, Yang et al. (2015) also revealed that the intense secondary formation of SNA contributed most of the formation of the hazes in October 2014, with the SOR and NOR values increasing considerably during the haze episodes. Similar variations were also observed in winter, but the formation of SNA in winter was much weaker than that in summer and autumn, with SOR values of only 0.18–0.35 and NOR values of 0.20–0.22 during the regional haze episode (Jan 13 to 15) at the three urban sites. However, the increases of the SNA concentrations were still significant during this haze episode. In addition, the increase in OM was pronounced during the winter regional haze episode, exhibiting the highest OM values during this episode with average values of 57.3, 51.8 and 133.4 $\mu g/m^3$ in Beijing, Tianjin and Shijiazhuang, respectively, which were 3.5, 4.1 and 5.9 times of those during the clean periods, while the enhancement ratio was only 1.1–1.4 in summer period. Therefore, SNA and OM both increased substantially and dominated the $PM_{2.5}$ during the winter haze episode, such that the respective contributions of SNA and OM were 42.6% and 26.1% in Beijing, 53.8% and 23.3% in Tianjin, and 43.2% and 37.0% in Shijiazhuang. The phenomenon of the significant increase of SNA and OM in winter haze episode was also observed by Zhao et al.(2013c) and Huang et al. (2014), indicating that the winter haze may be largely driven by secondary aerosol formation, which could be identified as a common characteristic of winter pollution in this region. Compared to winter, the SOR and NOR values increased on the whole in spring, but their variations, as well as the variations of SNA and OM, had no apparent regular connection with the fluctuation of $PM_{2.5}$, especially in Tianjin and Shijiazhuang. In contrast to those observed in summer and winter, the most striking feature of spring pollution was the enhanced contribution of mineral dust; during the regional dust-haze episode from Mar 29-D to 30-N, the contribution of the mineral dust even reached 60.3% in Shijiazhuang, 51.7% in Tianjin and 47.8% in Beijing on Mar 29-D under the conditions of a strong northerly wind (Fig. S6). Subsequently, the mineral dust decreased while SNA rose gradually with the increase of relative humidity and the shift of the wind direction (Fig. 8 and Fig. S6), and the contribution of SNA to $PM_{2.5}$ rose to 61.3%, 53.9% and 55.0% on Mar 30-N in Beijing, Tianjin and Shijiazhuang, respectively, again exhibiting secondary pollution characteristics.

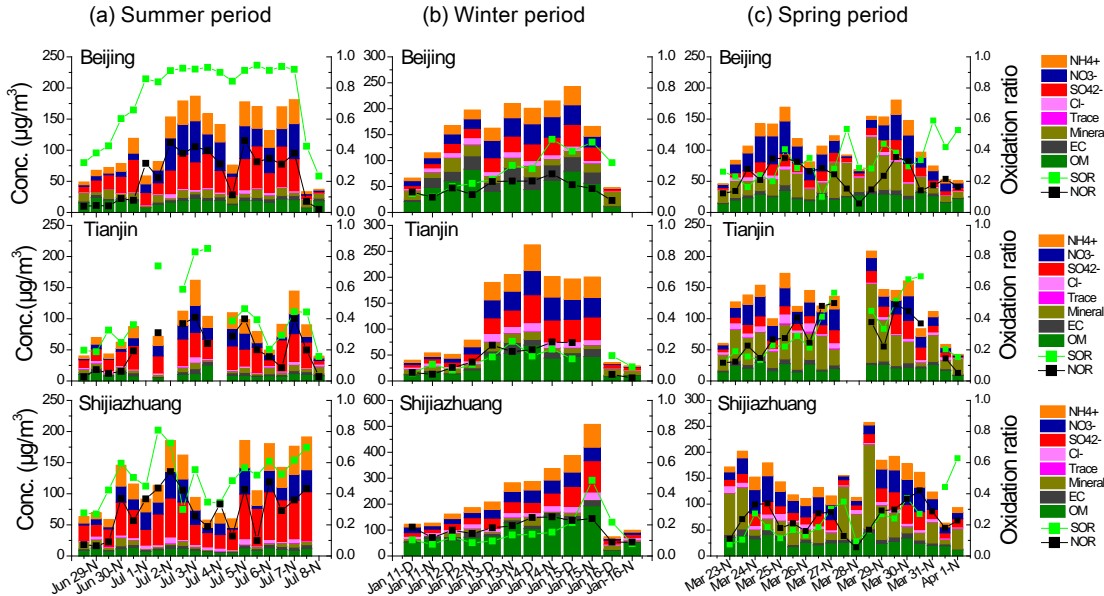

**Figure 8**. The evolutions of the chemical species, sulfur oxidation rate (SOR) and nitrogen oxidation rate (NOR) in the specific pollution periods in summer (a), winter (b) and spring (c) at the urban sites. D: daytime; N: nighttime

## 3.4.2 Evolution of source contributions

In addition to the evolutions of the chemical components during the transitions from clean to pollution processes, significant variations in the contributions of source/factor contribution were also observed and exhibited strong seasonal features and spatial heterogeneities (Fig. 9). However, the common characteristic of each season and site was that the secondary aerosol/inorganic aerosol played a key role in the development of haze pollution, which generally recorded increasing contributions with worsening pollution (except in spring of Shijiazhuang), which was also reported by Huang et al. (2014). Especially in summer and autumn, secondary inorganic aerosol increased most dramatically as a function of high relative humidity and suitable temperature at the urban sites; on heavy pollution days, it accounted for 55.7–75.2% of the PM$_{2.5}$ in summer and 55.0–61.5% in autumn, thus representing the biggest source of atmospheric PM$_{2.5}$ during these two seasons. In contrast, in Xinglong, the secondary aerosol source was always the dominant factor of the haze formation, which accounted for 66.7%, 67.5%, 68.4% and 87.0% of the PM$_{2.5}$ on heavy pollution days in summer, autumn, winter and spring, respectively. In addition, biomass burning was also another important source during the winter pollution process in Xinglong.

In contrast to background site, the emission sources and generation mechanisms of haze pollution were more complex at the urban sites, especially in winter, as the primary emissions, such as the motor vehicle exhaust, coal combustion and industrial processes were also the main sources of heavy pollution in winter. As the main fuel for winter heating in northern China, coal

combustion was crucial to the heavy pollution in winter in urban areas, as its contribution increases with increasing pollution levels. In Tianjin and Shijiazhuang, this source contributed nearly 30% to the $PM_{2.5}$ on heavy pollution days and contributed even more when considering the secondary formation of gaseous precursors emitted by coal combustion. Moreover, the primary emissions of motor vehicles also exerted a remarkable impact on the winter haze pollution, accounting for 26.1% of $PM_{2.5}$ on heavy pollution days in Beijing, and accounting for more when the secondary conversion of gaseous pollutants in vehicle exhaust are considered, as nitrate-rich secondary aerosol increased from 2.9% on clean days to 19.5% on heavy pollution days during the winter period. On hazy days, low visibility could aggravate urban traffic congestion during rush hour, thus causing more pollutants to be emitted by motor vehicles operating in these conditions (Zhang et al., 2011). Therefore, the winter haze was mainly formed by local processes (local direct emissions and secondary transformation).

In spring, the effect of dust source was highlighted at the urban sites. Most notably, in Shijiazhuang, the mineral dust source significantly contributed to the aerosol pollution process, as its contribution to $PM_{2.5}$ continuously increased from 9.7% on clean days to 18.6% on moderate pollution days to 22.9% on heavy pollution days. However, along with the increase in pollution levels, the ratio of local road dust source to the overall dust source decreased from 81.6% (on clean days) to 50% (on pollution days), thus reflecting the significant impact of the long-range transport of the northwest dust on the spring aerosol pollution in Shijiazhuang. Differing from the increase of relative humidity during haze episodes of the other seasons, the relative humidity decreased continuously from 64% (clean days) to 47% (moderate pollution days) and to 40% (heavy pollution days) in spring in Shijiazhuang (Table S2). Therefore, the heterogeneous reactions promoted by enhanced water vapor may not be the spring haze formation mechanism in Shijiazhuang, as the contribution of secondary inorganic aerosol decreased remarkably during the haze episode (Fig. 9 and Fig. S5). In contrast, the wind speed and the contribution of the dust source significantly increased during the spring haze period in Shijiazhuang. A similar but much milder pattern (except in terms of wind speed) was also recorded in Tianjin. Therefore, dust pollution, which was mainly the result of long-range transported soil dust and local road dust, contributed to the spring haze in Shijiazhuang and Tianjin.

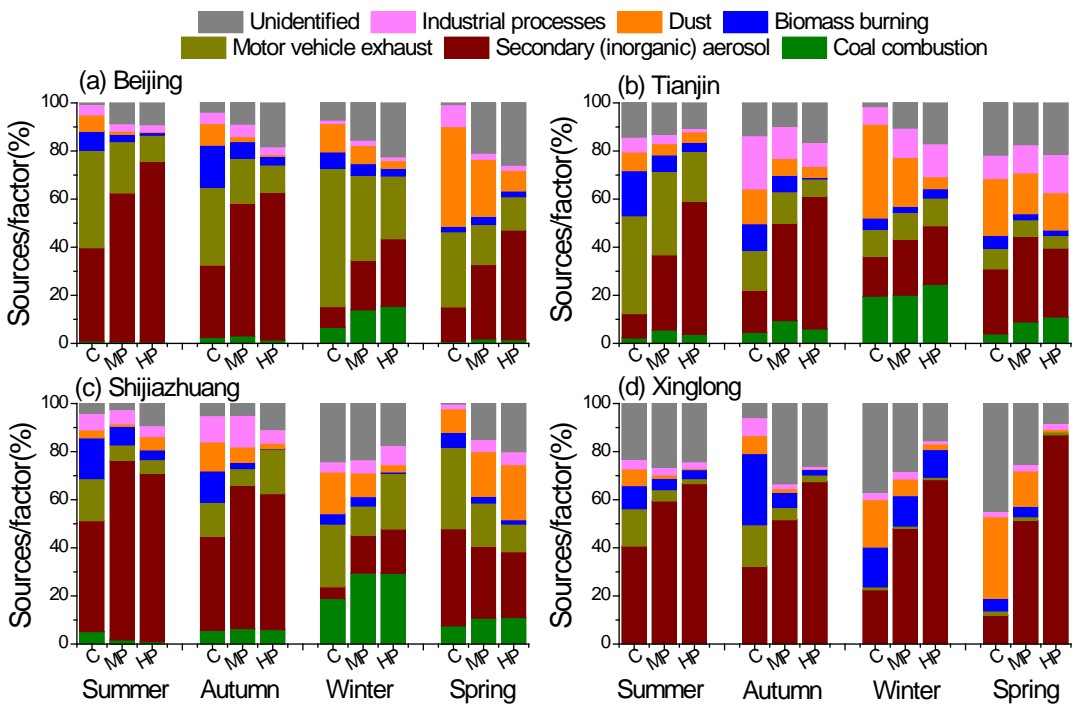

**Figure 9**. Fractional contributions of sources/factors to the PM$_{2.5}$ masses at different pollution levels during each season in Beijing (a), Tianjin (b), Shijiazhuang (c) and Xinglong (d). C, MP, and HP represent clean days (PM$_{2.5}$＜75 μg/m$^3$), moderate pollution days (75≤PM$_{2.5}$＜150 μg/m$^3$) and heavy pollution days (PM$_{2.5}$≥150 μg/m$^3$), respectively.

## 3.5 Backward trajectory analysis

To reveal the pollution patterns and source signals of the PM$_{2.5}$ carried by air masses from different directions and regions, the source contributions of PM$_{2.5}$ were grouped according to their trajectory clusters, as shown in Fig. 10. The results in Fig. 10 indicate the important effects of regional transport on PM$_{2.5}$. More than half of the air masses (54%, 64%, 51% and 56% for Beijing, Tianjin, Shijiazhuang and Xinglong, respectively) of the entire study period were from the BTH region and Shandong Province. These air masses, which move at slow speeds and at low heights could have carried abundant atmospheric pollutants (i.e., particles and gaseous pollutants) from the areas through which they passed, which may have been accompanied by plenty of water vapor during the transport process (Tao et al., 2012; Zhu et al., 2016), resulting in high PM$_{2.5}$ mass concentrations driven by local secondary formations at the sampling sites. The air masses (cluster 1 at each site) from the southern direction caused the most serious pollution. Air masses originating from Mongolia were also dominant (27–46%) in this region (cluster 2–3 in Beijing, cluster 4–5 at Tianjin, cluster 3–4 at Shijiazhuang and cluster 3 in Xinglong), and the PM$_{2.5}$ in these clusters were generally lower than those from the surrounding polluted areas, except for Shijiazhuang and Tianjin in cluster 5 (Fig. 10b). In addition, a small proportion of air masses originating from the Hulunbuir prairie in Inner Mongolia, such as those in cluster 3 in Tianjin and

in cluster 4 in Xinglong, could have carried clean air to the sampling sites, thus causing the corresponding $PM_{2.5}$ mass concentrations to be the lowest. In contrast to the other sites, the highest average $PM_{2.5}$ concentrations of each cluster were observed in Shijiazhuang, especially in cluster 2, which originated from Inner Mongolia and passed over Shanxi Province before arriving at the sampling sites and corresponded to the highest $PM_{2.5}$ value (178.9 μg/m$^3$); although cluster 4 originated from Mongolia and travelled at fast speeds and higher heights, the $PM_{2.5}$ was indeed higher than that of cluster 1, which may be because it passed over Inner Mongolia and Shanxi Province and thus could have carried many pollutants from these polluted areas; the $PM_{2.5}$ concentration in cluster 3, which originated from Mongolia and passed over Inner Mongolia and northern Hebei (a relatively clean area in the BTH region), was relatively lower (102.1 μg/m$^3$). However, the heavy pollution in Shijiazhuang was mainly dominated by cluster 1 (51%) from the south, as cluster 2 and cluster 4 accounted for only 23% and 13% of the trajectories, respectively. Similarly, the haze pollution in Beijing, Tianjin and Xinglong also developed due to the presence of weak southerly air masses from heavily polluted regions. This is consistent with the results of Guo et al. (2014) and Li et al. (2015).

In addition to the different $PM_{2.5}$ concentrations in the different clusters, considerable differences in the source contributions were also found. For example, in Shijiazhuang, high $PM_{2.5}$ concentrations were observed in each cluster. However, it could also be clearly seen that the source contribution charts of these clusters were very different and that the air masses originating from the BTH region and Shandong Province were characterized by high contribution from secondary inorganic aerosol, while the air masses from the long-range transports were more enriched in dust. Therefore, in Shijiazhuang, the contribution of secondary inorganic aerosol occurred such that cluster 1> cluster 2> cluster 3> cluster 4, whereas that of the dust source exhibited the opposite pattern. Similar patterns were also observed for Xinglong, Beijing and Tianjin in this study. This pattern of secondary inorganic aerosol was also observed by Zhang et al. (2014) in their study in Beijing. As mentioned in Section 3.3, the secondary aerosol was primarily attributed to the transformations of their precursors ($SO_2$, $NO_x$, $NH_3$ and VOCs). The slow and near-ground air masses originating from the regional polluted areas could have resulted in stagnant conditions, which could have been conducive to the accumulation of precursors from local emissions and those transported in and to the following secondary transformation. Furthermore, during this transportation, the carried gaseous pollutants could also have undergone secondary transformations and directly resulted in rapid increases in $PM_{2.5}$ concentrations in the downwind area (Bressi et al., 2014; Li et al., 2015). Our previous study also revealed that the high concentrations of organic aerosols (OA) in Beijing, especially those low-volatility oxygenated aerosols that are more oxidized and aged, were associated with southerly originating air masses containing secondary regional pollutants (Zhang et al., 2015b).

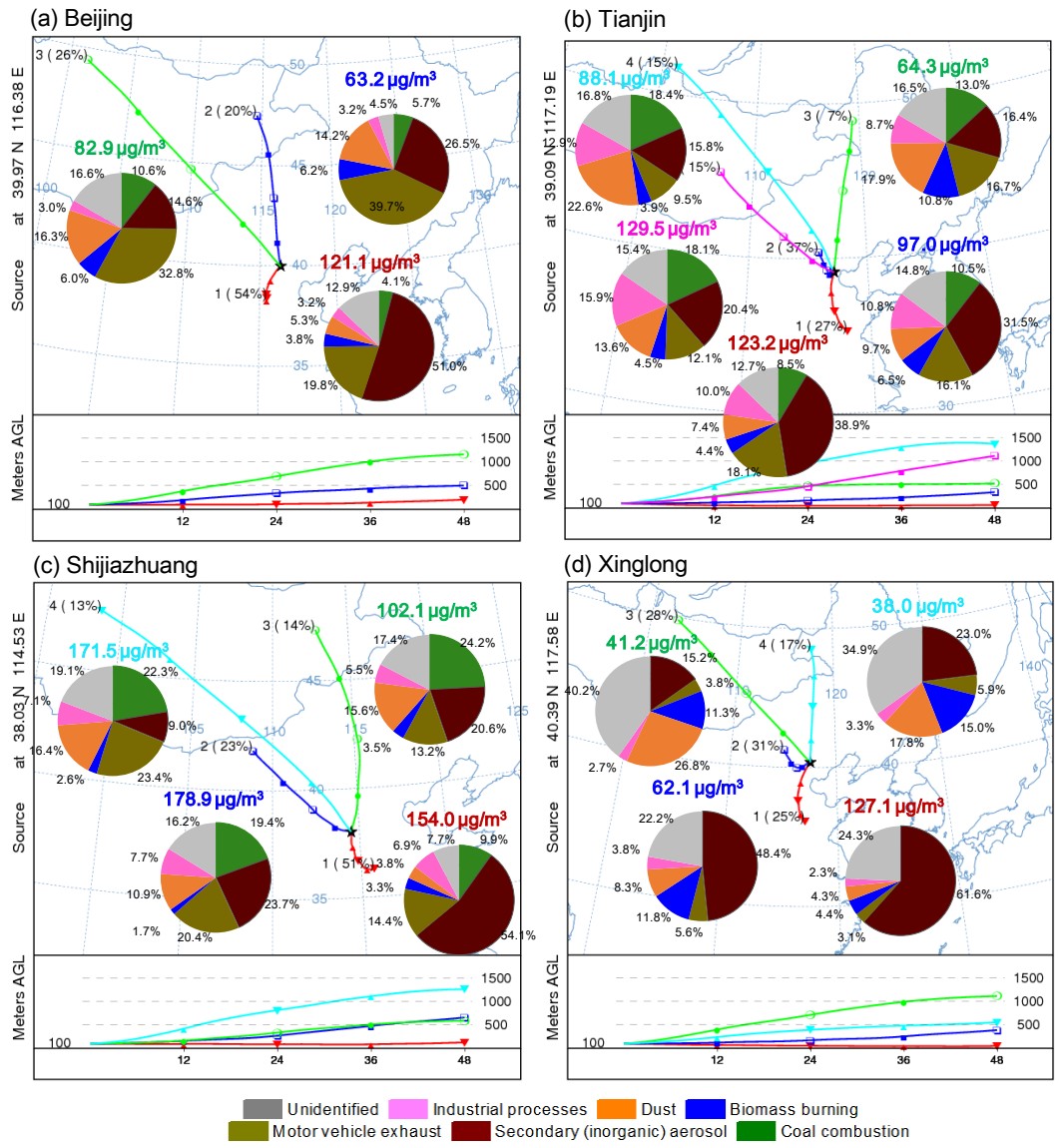

**Figure 10**. Source contributions resolved from the PMF at each 48 h backward trajectory cluster during the entire study period in Beijing (a), Tianjin (b), Shijiazhuang (c) and Xinglong (d)

## 4. Conclusions

5     In this study, the chemical compositions and emission sources of fine particulate matter (PM$_{2.5}$) were comprehensively investigated at three urban sites (Beijing, Tianjin and Shijiazhuang) and a background site (Xinglong) in the BTH region. Severe PM$_{2.5}$ pollution was found at all three urban sites, especially in Shijiazhuang, and the background site was found to be relatively clean. The seasonal variations of the PM$_{2.5}$ concentrations in Xinglong were not significant due to the

10     presence of fewer anthropogenic emissions, while at the urban sites, the lowest PM$_{2.5}$ values were observed in summer and the highest values were observed in winter, likely due to the prevalence of winter coal-fired heating and unfavorable meteorological conditions. The chemical compositions of PM$_{2.5}$ were similar at the four sites, and the major chemical components were organic matter, secondary inorganic ions (sulfate, nitrate, ammonium) and mineral dust. These

components showed distinctive seasonal variability, which were closely related to chemical processes, emission sources and meteorological conditions. Organic matter and elemental carbon had the highest recorded values and contributions to $PM_{2.5}$ in winter, sulfate peaked in summer, while nitrate peaked in autumn, and mineral dust peaked in spring.

The PMF model-resolved source analysis showed that coal combustion, motor vehicle exhaust, secondary inorganic aerosol, dust and industrial processes were the main sources of $PM_{2.5}$ in urban areas; however, the dominant source at the background site was the secondary aerosol. The drastic secondary transformation of gas precursors was the dominant cause of aerosol pollution, especially in summer and autumn. In winter, local direct emissions (coal combustion and motor vehicle exhaust) and secondary formations greatly impacted the haze formation in urban areas; while in spring, dust source exerted a significant impact and dominated the pollution in Shijiazhuang and Tianjin. In urban atmospheres, especially in Beijing, the contribution of motor vehicle exhaust was also prominent in haze formation, as it is the major source of gaseous $NO_x$. However, in this study, we could not determine the exact contribution of the secondary transformation of the $NO_x$ emitted by motor vehicles. Future studies should investigate the additional details about these secondary aerosols.

Haze pollution has remarkable regional characteristics, and the severe pollution in the BTH region was mainly influenced by the region itself and its surrounding polluted areas to the south. Therefore, we question the efficiency of the abatement strategies of emission reductions and air quality improvement and suggest a joint collaboration of cities in this region, or even throughout all of northern China. The reductions of gaseous precursors from fossil fuel combustion, which equates to the reduction of emissions from motor vehicles in Beijing probably by improving oil quality, and those from coal combustion from Tianjin, Hebei and the surrounding heavily polluted provinces, are essential to mitigate the severe haze pollution in BTH region.

## Acknowledgements:

This study was supported by the Ministry of Science and Technology of China (No. 2016YFC0202700) and the "Strategic Priority Research Program" of the Chinese Academy of Sciences (XDB05020000). The authors acknowledge the NOAA Air Resource Laboratory for its unrestricted provision of the HYSPLIT trajectory model. We are also thankful to the China Meteorological Administration for the meteorological data from Tianjin and Shijiazhuang and to Prof. Jianhui Bai from the Institute of Atmospheric Physics, Chinese Academy of Sciences for the meteorological data in Xinglong.

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
