# Peer review of "Chemical characterization and synergetic source apportionment of"

_Atmospheric Chemistry and Physics, 2017_

## Referee Comment (RC1) · Anonymous Referee #2 · 4 Jul 2017

This paper describe the chemical characterization and source apportionment of PM2.5 at four sites in the Beijing-Tianjin-Hebei (BTH) region, China. The topic of the paper is well suited for ACP, and the data itself are interesting. On the whole, the manuscript needs editing concerning the grammar and syntax by native English speaker. In addition, the manuscript suffers from many unclear statements. I have many points where

more information is needed or where I disagree.

Specific comments: Page 1, Line 20-34: The abstract could be improved greatly. The highlighted results in the abstract are not really exciting. I suggest the authors focus on the new findings on haze formation mechanism and the influence of regional transport.

Page 1, Line 32: Please define BTH at the first appearance. Abbreviations and acronyms are typically defined the first time they are used within the main text and then used throughout the remainder of the manuscript.

Page 1, Line 33-34: PMF was employed to apportion the source contribution to PM2.5. The sources of gaseous precursors are identified in this study?

Page 3, Lines 2-36: This part is not well written. The authors state that the studies on haze pollution in the BTH region have obtained fruitful and meaningful results. Please summarize the results on the chemical speciation, haze formation mechanism, emission sources, and influences of regional transport in North China. Meanwhile, the authors need to logically address why it is particularly important to do this study. The temporal and spatial characteristics can not be obtained based on single-site and short-term study. However, we still could get information from a lot of previous studies. What science questions need more research, particularly based on multiple-site and long-term study? Please clarify them.

Page 4, Lines 4-6: The authors state that they emphatically analyzed the chemical compositions and emission sources at different levels and the influence of air masses. Please highlight the new results or findings on them in the abstract.

Page 4, Line 14: "These site reflect the atmospheric pollutions condition in this region"? Reword this sentence. Please clarify the sampling strategy.

Page 4, Lines 27-28: Please clarify the distance between the filter sampling site and meteorological monitoring station in Tianjin, Shijiazhuang and Xinglong.

Page 7, Line 11: The calculation of mineral dust was performed on the basis of six

crustal element oxides. Why are Na, Mg, Zn not included in the Mineral dust? The calculations of Si, K and Fe are based on their ratios to Al in crustal dust. References should be added here.

Page 8, Lines 26-29: The authors state that the cycles of haze episodes are primarily driven by fluctuations in meteorological conditions such as wind speed, relative humidity, air temperature, atmospheric stability, the height of the planetary boundary layer and air mass origins. Please show the temporal trend of these meteorological parameters in 4-7 days here or in Supporting information. In my opinion, these parameters have more effect on the diurnal variation of air pollutants rather than the 4-7 days' cycle.

Page 9, Lines 26-35: The extreme pollution events on Oct 5-11 and Jan 2-6, Jan 11-16 are worth a in-depth discussion.

Page 10, Line 9: Shijiazhuang recorded high relative humidity and low wind speeds. It is usual or not from the historical record. Please clarify.

Page 11, Lines 10-21: Please provide the standard deviation of the ratio of NO3-/SO42in four sampling sites. The data reported here is annual average value? Please clarify.

Page 12, Lines 16-17: The authors state that strengthened burning activities may occur at night because of the higher night/day ratios of EC and CI- than that of PM2.5. Actually the photochemical reaction of secondary species and the boundary layer variation also could result in the higher night/day ratio of primary PM. Please provide more evidences to support this statement.

Page 12, Lines 29-31: Please provide the meteorological parameters at day and night in the four sites in the supporting information to support this statement.

Page 15: I still think the analysis on the whole pollution processes particularly the extreme pollution events could get more information on haze formation mechanism than that of the different pollution levels.

Page 15, Line 32: The OC/EC ratio increases with the increasing development of haze

pollution? It is different from the statement in Lines 19-20. Is there any study on the secondary organic carbon in the wintertime of Beijing and Shijiazhuang? Maybe the sources affect the OC/EC ratio in different pollution levels. Again, the discussion on the specific pollution process will avoid this bias associated with emission sources.

Page 16, Lines 22-25: I strongly suggest to discuss he differences between the spring haze and winter haze. The formation mechanism may be totally different.

Page 21, Lines 19-page 22, Line 19: The authors should consider incorporating this discussion into the other sections of the manuscript so that comparisons are made when results are discussed.

СЗ

---

## Referee Comment (RC2) · Anonymous Referee #1 · 4 Jul 2017

In this study, atmospheric PM2.5 samples were collected for one year at three urban sites and one regional background site in the BTH and analyzed for their chemical compositions. Emission sources of PM2.5 at the four sites were comprehensively investigated by using PMF and backward trajectory analysis. Some important findings were reported, which is helpful for readers to improve their understanding on the pollution situation in the region. Within this reviewer's knowledge, this work may be the first study using unified approach for sampling and subsequent analysis to perform synergetic source apportionment of PM2.5 in BTH region. In general, this paper was

well written, and the scientific contents fall in the scope and interest of the journal of ACP. Therefore, this referee recommends it to be accepted for final publication in Atmospheric Chemistry and Physics with minor revisions after addressing following questions.

General comments 1. The applied analysis method of PMF must be explained in more detail. How the authors prepared the error matrix and especially how they dealt with combining errors from different measurement techniques. In addition, the authors do not discuss how many solutions they explored (e.g. 1-10 solutions), and how similar or different the resulting factors are. 2. Eight factors were identified for Beijing and Tianjin, while nine were identified for Shijiazhuang and only five were identified for Xinglong. Although the first six factors were well identified, the remaining factors were poor discussed. What are the differences of these factors among the three megacities? Furthermore, the source apportionment results of the presented manuscript should be compared with the previous studies. 3. It would be more concise and easy to follow if the authors could condense the manuscript by removing the Section 3.2.2 to the Supplement, as the results of "Diurnal variation" is not essential to the expression of main points of this manuscript.

Specific comments: 1. Page 1, line 31-34. The conclusion of improving the quality of oil products from motor vehicles is weird. As haze pollution usually occurs when air masses originated from polluted industrial regions of the south prevailed, the control strategies should be focused on fossil fuel combustion, like coal combustion. 2. Page 2, line4: The first document named as "Zhang et al., 2015b" should be "Zhang et al., 2015a". 3. Page2, line 20: Please briefly discuss the relationship between chemical components and health effects, and add more results about the effect of chemical components on the environment and climate. 4. Page 3, line 23: "have reported that a new ...." should be "have reported a new ...." 5. Page 3, line 34: "...., these studies yield a narrow view of their ....", here, "their" is ambiguous. 6. Page 8, line 1: You should introduce the eight carbon fractions in Section 2.2.2 firstly. Otherwise, we don't
know what are OC1, OC2, OC3..... 7. Page 8, line 30: The values of PM2.5 annual average concentration should be shown as "mean $\pm$  standard deviation" 8. Page 9, Figure 2: Please add an instruction about BJ, TJ, SJZ and XL 9. Page 10 line 2 to 5, I think the frequent rain is also very important for the PM clear in summer. 10. Page 11, line 15: "Therefore, the NO3ïij=/SO42ïij=mass ratio was larger than 1.0 at Beijing, implying that the predominance of motor vehicle emissions over coal combustion in the contribution to PM pollution" this statement implies that motor vehicle emissions is the single source of NOx , which is incorrect, as coal combustion from power plants is another important source. 11. Page 12, Section 3.2.2: The title should not be "diurnal variation". but "day-night variation". 12. Page 12, line 26: "(Sun, et al., 2016)" should be "(Sun et al., 2016)" 13. Page 15, line 15: "noticeably A remarkable....." should be "noticeably. A remarkable.....". 14. Page 15, line18-35: The authors discussed about the variation of OC/EC mass ratio with the pollution level, would you please display as a chart to make it more intuitive? 15. Page 15 line 31. "Therefore, the fact that the OC/EC ratio increases with the increasing development of haze pollution....." This statement is wrong based on the context, it should be " ... the OC/EC ratio decreases with the increasing ... "16. Page 15, Section 3.2.4: I suggest the authors show the meteorological parameters at different pollution level in the Supplement to further support the analysis of accumulation and enhanced secondary formation on pollution days. 17. Page 22, Figure 9: Please add an instruction about "C, MP, HP" 18. Page 25, Section 3.4. Pease add more information about the backward trajectory analysis. 19. Page 25, line 6ïijŽ"in three urban sites" should be "at ... sites" 20. Page 25, Figure 10: I believe the font in this figure is inconsistent. Please revise that. 21. Page 26, line 1-6: This part should be improved with a brief description of the chemical composition while mainly focusing on the seasonal characteristics of major components.

---

## Author Comment (AC1) · 15 Sep 2017

Dear editor and referees,

We appreciate the referees for very careful reading and valuable comments, which greatly contribute to improving the quality of this paper. We take these comments very seriously, modified the manuscript accordingly and replied in our point-by-point response attached below. In addition, the manuscript concerning the grammar and syntax has been improved by native English speaker. The response is highlighted in blue and changes in the manuscript are colored in red.

**Response to Anonymous Referee #1**

Anonymous Referee #1

In this study, atmospheric PM2.5 samples were collected for one year at three urban sites and one regional background site in the BTH and analyzed for their chemical compositions. Emission sources of PM2.5 at the four sites were comprehensively investigated by using PMF and backward trajectory analysis. Some important findings were reported, which is helpful for readers to improve their understanding on the pollution situation in the region. Within this reviewer's knowledge, this work may be the first study using unified approach for sampling and subsequent analysis to perform synergetic source apportionment of PM2.5 in BTH region. In general, this paper was well written, and the scientific contents fall in the scope and interest of the journal of ACP. Therefore, this referee recommends it to be accepted for final publication in Atmospheric Chemistry and Physics with minor revisions after addressing following questions.

General comments

1. The applied analysis method of PMF must be explained in more detail. How the authors prepared the error matrix and especially how they dealt with combining errors from different measurement techniques. In addition, the authors do not discuss how many solutions they explored (e.g. 1-10 solutions), and how similar or different the resulting factors are.

Response: The consideration of preparation error matrix for PMF are mainly based on the analytical error for each chemical species as the accuracy of the measurement techniques and analysis is different for different species. The particulate sulfate, nitrate and ammonium were abundant in $PM_{2.5}$ and their determination were with good quality control, then the error fractions for these three components were estimated at only 5%. The laser reflection correction method was used for the determination of OPC (organic pyrolyzed carbon) and the precise determination of its concentration is influenced by many factors, and EC2 and EC3 were with low concentrations. Thus, OPC, EC2 and EC3 may have relatively high analytical error, then the their error fractions were estimated at 15%. And we estimated 10% error fractions for other species. In addition, we made OPC, EC2, EC3 and $Mg^{2+}$ in Beijing, OPC in Shijiazhuang, and Cr in Xinglong a weak category since they had relatively low signal-to-noise ratios (S/N), and there were no excluded species at all four sites. Moreover, PMF analysis requires a complete data set, the samples with missing values of individual species were all excluded in order to reduce the error, rather than replaced by the geometric mean of the remaining observations.

In this study, we explored 8 PMF solutions (5−12 factors) at the urban sites and 7 solutions (3−9 factors) at the regional background site. The factor number was determined based on the interpretability of different PMF solutions as well stability across bootstrap-replicate data sets as

represented by factor matching rate (Xie et al., 2013a, 2013b).

2. Eight factors were identified for Beijing and Tianjin, while nine were identified for Shijiazhuang and only five were identified for Xinglong. Although the first six factors were well identified, the remaining factors were poor discussed. What are the differences of these factors among the three megacities? Furthermore, the source apportionment results of the presented manuscript should be compared with the previous studies.

Response: In this study, we have used united method to resolve the source apportionment at all four sites and have obtained different factors at each site (eight for Beijing and Tianjin, nine for Shijiazhuang and five for Xinglong) . To facilitate the discussion, we classified these factors into six categories, such as coal combustion, secondary aerosol/inorganic aerosol, motor vehicle emissions, biomass burning, mineral dust and industrial processes. Meanwhile, some categories may contain more than one factor, such as the category of secondary aerosol/inorganic aerosol includes secondary aerosol, secondary inorganic aerosol, or the sum of nitrate-rich secondary and sulfate-rich secondary aerosol; the category of mineral dust contains road dust and soil dust. Therefore, the listed six factors in the manuscript covers all the factors that resolved at all four sites.

The identified $PM_{2.5}$ sources of this study have been compared with those of recently published studies and the results from the Beijing Municipal Environmental Protection Bureau (http://www.bjepb.gov.cn/), the Tianjin Municipal Environmental Protection Bureau (http://www.tjhb.gov.cn/) and the Shijiazhuang Municipal Environmental Protection Bureau (http://www.sjzhb.gov.cn/), which are shown in Table S3 in the Supplement. And related discussion has been added in the manuscript (Page 19, line 5–17 of the manuscript), as follows:

" In summary, secondary inorganic aerosol (40.5%) and motor vehicle exhaust (24.9%) were the largest $PM_{2.5}$ sources in Beijing and have greater values than the results of the 2010 study by Wu et al. (2014). The contribution of motor vehicle exhaust was close to that provided by the Beijing Municipal Environmental Protection Bureau (19.9–22.4%, http://www.bjepb.gov.cn/) (Table S3). Compared with those of Beijing, there were more complicated sources of $PM_{2.5}$ in Tianjin and Shijiazhuang. Motor vehicle exhaust was also an important source at the two sites, but the contribution (15.2% in Tianjin and 17.3% in Shijiazhuang)was lower than that in Beijing, which was consistent with the results published by the Environmental Protection Bureau (EPB) (Table S3). However, coal combustion became an important source in Tianjin (12.4%) and Shijiazhuang (15.5%), which was slightly lower than that from the Environmental Protection Bureau and that given by Liu et al. (2017a) (16.5% in Tianjin). The biggest $PM_{2.5}$ source in Tianjin (29.2%) and Shijiazhuang (36.4%) was still secondary inorganic aerosol, of which the contribution in Tianjin was close to that of Liu et al. (2017a) (26.1%)."

3. It would be more concise and easy to follow if the authors could condense the manuscript by removing the Section 3.2.2 to the Supplement, as the results of "Diurnal variation" is not essential to the expression of main points of this manuscript.

Response: we have removed this section to the Supplement.

**"Supplementary text for describing the day-night variations of $PM_{2.5}$ and major chemical components**

The analysis of the day-night variations indicates that the differences in the $PM_{2.5}$ annual average concentrations during the day and those during the night were significant at the urban sites, where the values were 8–19% higher at night than those during the day, while negligible differences were found on the annual scale in Xinglong (Fig. S7). This obvious day-night variation of the $PM_{2.5}$ concentrations in urban areas was probably due to the apparent changes in the height of the mixing layer between day and night (Zhao et al., 2009). However, in Xinglong, the dominant source was from the regional or long-range transport, with fewer contributions from local emissions; thus, the nocturnal stable boundary layer could have reduced the quantity of transmissions from the outside. The chemical compositions also recorded obvious day-night variations, as the mass ratio of $NO_3^-/SO_4^{2-}$ recorded higher values at night (0.99–1.39) than during the day (0.81–1.13), which is consistent with the similar results obtained by Sun et al. (2016) in Xianghe, which is located approximately 50 km southeast of Beijing. Such day-night variations indicate the important role of the gas-phase photochemical production of sulfate during the day while the facilitated gas-to-particle partitioning of semi-volatile nitrate is associated with the low temperatures (Sun et al., 2016) and effective hydrolysis of $N_2O_5$ at night, which is a major source of nitric acid in the urban atmosphere during the night and is more efficient on wet surfaces (Zhang et al., 2015). In addition, the relatively static and stable meteorological conditions at night resulted in obviously lower fractions of mineral dust (11.3–17.0%, except for in Tianjin), than those recorded during the day (18.3–24.3%)."

[Figure]

Figure S7. Day-night variations of $PM_{2.5}$ and major chemical components, based on annual data

Specific comments:

1. Page 1, line 31-34. The conclusion of improving the quality of oil products from motor vehicles is weird. As haze pollution usually occurs when air masses originated from polluted industrial regions of the south prevailed, the control strategies should be focused on fossil fuel combustion, like coal combustion.

Response: It is true that haze pollution in Beijing usually occurs when air masses originating from polluted industrial regions to the south prevailed. However, this usually occurs during the initial stage of haze episodes and acts as an inducement for the occurrence of haze episodes (Wang et al., 2017), while during the peak stage of haze episodes, particularly extremely heavy haze episodes, the pollutions are often local-dominated. In this study, with the development of haze process, the

concentration and contribution of nitrate recorded a pronounced increase during the pollution process and the OC/EC mass ratio showed a significant reduction, both reflected the dominant contribution of local motor vehicle exhaust in haze episodes. Therefore, controlling local emissions is a much more important measure to alleviate the extreme haze episodes in Beijing (Wang et al., 2017), we should focus on the controlling the quantity of motor vehicles and improving the oil product for Beijing. Of course, emission control in surrounded pollution areas, especially in Shijiazhuang, Tianjin, Tangshan, Baoding, Langfang and Cangzhou, as well as Henan and Shandong province, are also important to reduce the $PM_{2.5}$ concentrations and the occurrence of haze episodes in Beijing (Wang et al., 2017). Therefore, we have improved this presentation to be more exact as follows (Page 1, line 33–36) :

" This study suggests that the control strategies to mitigate haze pollution in the BTH region should be focused on the reduction of gaseous precursor emissions from fossil fuel combustion (motor vehicle emissions in Beijing and coal combustion in Tianjin, Hebei and nearby provinces)."

2. Page2, line4: The first document named as "Zhang et al., 2015b" should be "Zhang et al., 2015a".
Response: It has been revised now.

3. Page2, line 20: Please briefly discuss the relationship between chemical components and health effects, and add more results about the effect of chemical components on the environment and climate.
Response: We have revised this part as follows (Page2, line 21–33):

"Haze pollution with high fine PM loading could profoundly impact ecosystems, regional-scale atmospheric visibility, traffic safety, the economy, and interactions with climate (Zhang et al., 2015a); more importantly, this pollution can have adverse effects on human health, including the increased risks of respiratory, cardiac and other medical conditions (Elliot et al., 2016; Wu et al., 2017), thus leading to increased mortality rates, especially in megacities, which are generally seriously polluted and densely populated. In addition to the particle mass concentration and particle size, the health effects of PM are closely related to its chemical composition (Zhang et al., 2015a), and different diseases respond differently to different air pollutants (Tang et al., 2017). Moreover, the climate and environmental domino effects of PM are also closely related to the PM chemical compositions due to their different optical properties, such as those of black carbon, mineral particles, and brown carbon, which are light absorbing, while organic matter, ammonium sulfate and ammonium nitrate are light scattering (Tao et al., 2014; Wang et al., 2015; Wu et al., 2009; Zhang et al., 2016). "

4. Page 3, line 23: "have reported that a new ...." should be "have reported a new ...."
Response: It has been revised in the manuscript.

5. Page 3, line 34: "....., these studies yield a narrow view of their ....", here, "their" is ambiguous.
Response: we have revised this presentation.

6. Page 8, line 1: You should introduce the eight carbon fractions in Section 2.2.2 firstly. Otherwise, we don't know what are OC1, OC2, OC3......

Response: The introduction of the eight carbon fractions has been added in the part of Section 2.2.2 now in Page 6, line 23–28, as follows:

" In a pure helium atmosphere, OC1, OC2, OC3 and OC4 are produced stepwise at 140 ℃, 280 ℃, 480 ℃ and 580 ℃, respectively; followed by EC1 (540 ℃), EC2 (780 ℃) and EC3 (840 ℃) in a 2% oxygen-contained helium atmosphere. The OPC (organic pyrolyzed carbon) is determined when the reflected laser signal returns to its initial value after oxygen is added to the analyzed atmosphere. Therefore, the OC is operationally defined as OC=OC1+OC2+OC3+OC4+OPC, while the EC is defined as EC=EC1+EC2+EC3–OPC. Detailed procedures can be found in Xin et al. (2015)."

7. Page 8, line 30: The values of PM2.5 annual average concentration should be shown as "mean±standard deviation"

Response: The standard deviation has been added now.

8. Page 9, Figure 2: Please add an instruction about BJ, TJ, SJZ and XL

Response: To be consistent with other figures, the abbreviations of the sampling sites in Figure 2 have been replaced with their full names as follows:

[Figure]

9. Page 10 line 2 to 5, I think the frequent rain is also very important for the PM clear in summer.

Response: We quite agree with this. Rain is very beneficial to the wet deposition of atmospheric particulate matter. And summer is characterized by frequent rain usually accounting for 75% of annual rainfall in Beijing (Zhang et al., 2013) and is accompanied by great rainfall intensity. However, the data of precipitation during the study period were not available to us, thus we did not discuss its impact on the wet deposition of PM.

10. Page 11, line 15: "Therefore, the NO3- /SO42- mass ratio was larger than 1.0 at Beijing, implying that the predominance of motor vehicle emissions over coal combustion in the contribution to PM pollution" this statement implies that motor vehicle emissions is the single source of NOx ,which is incorrect, as coal combustion from power plants is another important source.

Response: Actually, Zhao et al. (2013a) reported that atmospheric NOx in China are mainly emitted from power plants, industry and transportation in 2010. Although many previous studies have shown that coal combustion emissions from power plants are also the most important sources of NOx in China, vehicular emissions contribute much more to surface NOx concentrations than power plants due to the elevated heights of emission stacks (Pan et al., 2016), and the widespread use of pollution control devices on power plants (fitted with NOx removal systems, such as the Selective Catalytic Reduction system) has greatly decreased the emission of NOx under the "Twelfth Five-Year Plan for National Environment Protection" in China (Gu et al., 2013; Zong et al., 2017). In contrast, the explosive growth in car ownership in recent years has resulted in vehicle exhaust becoming an important source of NOx pollution (Zong et al., 2017), especially in Beijing and Tianjin, the emission intensity of NOx is very strong due to numerous automobiles accumulate in small areas (Zhao et al., 2012). And atmospheric $SO_2$ are mainly from coal combustion emissions (83% in the Huabei region) (Zhao et al., 2012). Therefore, the mass ratio of $NO_3^-/SO_4^{2-}$ could reflect the relative contribution of mobile and stationary sources to the aerosol pollution to a certain extent. However, the statement of "Therefore, the $NO_3^-/SO_4^{2-}$ mass ratio was larger than 1.0 at Beijing, implying that the predominance of motor vehicle emissions over coal combustion in the contribution to PM pollution" in the manuscript is lack of rigor, we have revised it as:

" Therefore, the higher $NO_3^-/SO_4^{2-}$ mass ratio of Beijing implies that the predominance of motor vehicle emissions in the contributions to PM pollution (Han et al., 2016; Yang et al., 2015), while in Tianjin and Shijiazhuang, coal combustion may still play a dominant role. ".

11. Page 12, Section 3.2.2: The title should not be "diurnal variation", but "day-night variation".

Response: Thank the referee for pointing out this incorrect presentation, indeed, this section is "day-night variation" rather than "diurnal variation". Therefore, we have revised it now. In addition, we have removed this section to the Supplement to make the structure of this article more concise.

12. Page 12, line 26: "(Sun, et al., 2016)" should be "(Sun et al., 2016)"

Response: It has been revised in the manuscript.

13. Page 15, line 15: "noticeably A remarkable......" should be "noticeably. A remarkable......".

Response: It has been revised in the manuscript.

14. Page 15, line18-35: The authors discussed about the variation of OC/EC mass ratio with the pollution level, would you please display as a chart to make it more intuitive?

Response: Thanks for this good suggestion. The variation of OC/EC mass ratio with the pollution level has been added in this figure (now Fig. 7).

[Figure]

Figure 7. The evolutions of each aerosol chemical species and the ratio of OC/EC (marked with pink dots) at different pollution levels during the entire observational period. C, MP, and HP represent clean days ($PM_{2.5} < 75$ μg/m³), moderate pollution days ($75 \leq PM_{2.5} < 150$ μg/m³) and heavy pollution days ($PM_{2.5} \geq 150$ μg/m³), respectively. "%" represents the proportion of the filter sample quantity at each pollution level out of the total samples.

15. Page 15 line 31. "Therefore, the fact that the OC/EC ratio increases with the increasing development of haze pollution......" This statement is wrong based on the context, it should be " ... the OC/EC ratio decreases with the increasing ..."

Response: We are sorry for our carelessness. Actually, the OC/EC ratio in Beijing decreased with the increasing development of haze pollution. The wrong presentation has been revised now.

16. Page 15, Section 3.2.4: I suggest the authors show the meteorological parameters at different pollution level in the Supplement to further support the analysis of accumulation and enhanced secondary formation on pollution days.

Response: Thanks for this suggestion, we quite agree that the meteorological parameters at different pollution level and their day-night variations are necessary to assist the analysis of accumulation and enhanced secondary formations on pollution days. Therefore, we have shown the meteorological parameters at different pollution level and their day-night variations in Table S2 in the Supplement, and the time series of meteorological parameters during specific pollution periods in Beijing in Fig. S6 in the Supplement.

Table S2 The meteorological parameters under specific conditions during the sampling periods

|  |  | Temperature (℃) | | | | Relative humidity (%) | | | | Wind speed (m/s) | | | |
| --- | --- | --- | --- | --- | --- | --- | --- | --- | --- | --- | --- | --- | --- |
|  |  | Summer | Autumn | Winter | Spring | Summer | Autumn | Winter | Spring | Summer | Autumn | Winter | Spring |
| Beijing | Annual | 28.0 | 18.0 | 1.5 | 14.0 | 51 | 59 | 32 | 35 | 1.0 | 0.8 | 1.5 | 1.7 |
|  | C[a] | 28.3 | 18.6 | 1.5 | 12.1 | 44 | 46 | 21 | 32 | 1.2 | 1.2 | 2.3 | 2.3 |
|  | MP[b] | 27.4 | 18.7 | 2.3 | 14.2 | 57 | 69 | 34 | 33 | 0.9 | 0.6 | 1.4 | 1.6 |

| | | | | | | | | | | | | | |
|---|---|---|---|---|---|---|---|---|---|---|---|---|---|
| | HP[c] | 27.7 | 17.5 | 1.1 | 16.4 | 67 | 69 | 41 | 42 | 0.7 | 0.6 | 0.9 | 1.2 |
| | D[d] | 30.3 | 19.8 | 2.8 | 16.0 | 43 | 51 | 28 | 27 | 1.4 | 1.0 | 1.7 | 2.1 |
| | N[e] | 25.8 | 16.2 | 0.1 | 11.2 | 60 | 67 | 37 | 43 | 0.7 | 0.6 | 1.3 | 1.3 |
| Tianjin | Annual | 27.7 | 18.4 | 1.1 | 13.0 | 54 | 60 | 41 | 41 | 1.5 | 1.3 | 1.4 | 1.8 |
| | C | 28.6 | 18.9 | 1.5 | 11.2 | 47 | 61 | 31 | 39 | 1.6 | 1.4 | 1.8 | 2.1 |
| | MP | 27.2 | 19.1 | 1.1 | 13.3 | 59 | 61 | 40 | 44 | 1.4 | 1.1 | 1.6 | 1.8 |
| | HP | 28.4 | 16.9 | 1.7 | 13.4 | 66 | 53 | 46 | 40 | 1.2 | 1.2 | 1.1 | 1.5 |
| | D | 29.7 | 20.1 | 2.6 | 15.2 | 47 | 53 | 35 | 34 | 1.8 | 1.5 | 1.7 | 2.1 |
| | N | 25.7 | 16.6 | -0.4 | 10.9 | 62 | 69 | 45 | 48 | 1.2 | 1.0 | 1.2 | 1.5 |
| Shijiazhuang | Annual | 26.9 | 17.9 | 1.0 | 13.8 | 63 | 75 | 41 | 46 | 1.2 | 0.9 | 1.0 | 1.5 |
| | C | 28.6 | 16.3 | 1.9 | 9.1 | 50 | 64 | 21 | 64 | 1.3 | 1.1 | 1.3 | 1.2 |
| | MP | 26.6 | 18.9 | 1.6 | 13.0 | 66 | 78 | 35 | 47 | 1.2 | 0.9 | 1.4 | 1.5 |
| | HP | 26.2 | 17.7 | 0.6 | 15.5 | 71 | 78 | 46 | 40 | 1.1 | 0.7 | 0.9 | 1.6 |
| | D | 29.2 | 19.6 | 3.0 | 16.0 | 54 | 68 | 36 | 39 | 1.4 | 1.1 | 1.2 | 1.9 |
| | N | 24.8 | 16.2 | -1.0 | 11.5 | 70 | 83 | 47 | 53 | 1.0 | 0.7 | 0.8 | 1.2 |
| Xinglong | Annual | 20.6 | 12.1 | -4.8 | 7.0 | 67 | 66 | 40 | 44 | 2.3 | 2.5 | 2.7 | 3.3 |
| | C | 20.6 | 12.1 | -5.0 | 6.5 | 61 | 58 | 35 | 30 | 2.1 | 2.6 | 2.9 | 3.4 |
| | MP | 19.7 | 14.0 | -3.1 | 8.9 | 80 | 68 | 42 | 52 | 2.5 | 3.3 | 2.6 | 3.4 |
| | HP | 21.5 | 13.8 | -4.9 | 8.9 | 85 | 74 | 79 | 66 | 2.7 | 2.4 | 1.4 | 3.6 |
| | D | 21.7 | 13.2 | -3.7 | 8.3 | 64 | 62 | 36 | 38 | 2.4 | 2.7 | 2.7 | 3.6 |
| | N | 19.4 | 10.9 | -5.8 | 5.5 | 70 | 71 | 43 | 50 | 2.1 | 2.3 | 2.7 | 3.1 |

[a] Clean days (PM$_{2.5}$＜75 μg/m$^3$); [b] Moderate pollution days (75≤PM$_{2.5}$＜150 μg/m$^3$); [c] Heavy pollution days (PM$_{2.5}$≥150 μg/m$^3$);

[d] Daytime; [e] Nighttime

[Figure]

Figure S6. Time series variations of the meteorological parameters from Beijing during the specific pollution periods in summer (a), winter (b) and spring (c).

17. Page 22, Figure 9: Please add an instruction about "C, MP, HP"

Response: It has been added in the manuscript now.

18. Page 25, Section 3.4. Pease add more information about the backward trajectory analysis.

Response: It has been added in Section 2.4 in Page 9, line 3–10 , as follows:

" The backward trajectory analysis method is widely applied to identify the potential source regions and transport pathways of air masses, especially for serious air pollution episodes (Gao et al., 2015; Hu et al., 2012; Zhang et al., 2014). In this study, 48 h backward trajectories terminated at a height of 100 m above ground level were calculated for the all four sampling sites using the Hybrid Single-Particle Lagrangian Integrated Trajectory (HYSPLIT 4.9) model developed by the U.S. National Oceanic and Atmospheric Administration/Air Resources Laboratory (NOAA/ARL). The trajectories were calculated every 12 h, with starting times at 8:00 and 20:00 local time (corresponding to each sampling) during the entire observational period. "

19. Page 25,line 6:    "in three urban sites" should be "at ... sites"
Response: It has been revised in the manuscript.

20. Page 25, Figure 10: I believe the font in this figure is inconsistent. Please revise that.
Response: It has been revised as follows:

[Figure]

Figure 10. Source contributions resolved from the PMF at each 48 h backward trajectory cluster during the entire study period in Beijing (a), Tianjin (b), Shijiazhuang (c) and Xinglong (d)

21. Page 26, line 1-6: This part should be improved with a brief description of the chemical composition while mainly focusing on the seasonal characteristics of major components.

Response: It has been revised as follows (Page 27, line 5 to Page 28, line 4):

[revised manuscript text omitted]

**Response to Anonymous Referee #2**

Anonymous Referee #2
This paper describe the chemical characterization and source apportionment of PM2.5 at four sites in the Beijing-Tianjin-Hebei (BTH) region, China. The topic of the paper is well suited for ACP, and the data itself are interesting. On the whole, the manuscript needs editing concerning the grammar and syntax by native English speaker. In addition, the manuscript suffers from many unclear statements. I have many points where more information is needed or where I disagree.

Specific comments:

Page 1, Line 20-34: The abstract could be improved greatly. The highlighted results in the abstract are not really exciting. I suggest the authors focus on the new findings on haze formation mechanism and the influence of regional transport.

Response: Thanks for the good suggestion, we have improved the abstract greatly by deleting less important information and focusing on the meaningful results about source variations in seasons, locations and at different pollution levels, and on the influence of regional transport. The abstract has been revised as follows:

" The simultaneous observation and analysis of atmospheric fine particles ($PM_{2.5}$) on a regional scale is an important approach to develop control strategies for haze pollution. In this study, samples of filtered $PM_{2.5}$ were collected simultaneously at three urban sites (Beijing, Tianjin, and Shijiazhuang) and at a regional background site (Xinglong) in the Beijing-Tianjin-Hebei (BTH) region from June 2014 to April 2015. The $PM_{2.5}$ at the four sites mainly comprised organic matter, secondary inorganic ions (sulfate, nitrate and ammonium) and mineral dust. Positive matrix factorization (PMF) demonstrated that, on an annual basis, secondary inorganic aerosol was the largest $PM_{2.5}$ source in this region, accounting for 29.2–40.5% of the $PM_{2.5}$ mass at the urban sites; the second largest $PM_{2.5}$ source was primary emission of motor vehicle exhaust in Beijing (24.9%), whereas coal combustion in Tianjin (15.2%) and Shijiazhuang (17.3%), particularly in winter. Secondary inorganic aerosol play a vital role in the haze process, with the exception of the spring haze of Shijiazhuang and Tianjin, where the dust source was crucial. In addition to secondary transformations, local direct emissions (coal combustion and motor vehicle exhaust) significantly contribute to the winter haze at the urban sites. Moreover, with the aggravation of haze pollution, the OC/EC mass ratio of $PM_{2.5}$ decreased considerably and the nitrate-rich secondary aerosol increased during all four seasons in Beijing, both of which indicate that local motor vehicle emissions significantly contribute to the severe haze episodes of Beijing. To assess the impacts of regional transport on haze pollution, the PMF results were further processed with backward trajectory clusters analysis, revealing that haze pollution usually occurred when air masses originating from polluted industrial regions in the south prevailed and is characterized by high $PM_{2.5}$ loadings with considerable contributions from secondary aerosol. This study suggests that the control strategies to mitigate haze pollution in the BTH region should be focused on the reduction of gaseous precursor emissions from fossil fuel combustion (motor vehicle emissions in Beijing and coal combustion in Tianjin, Hebei and nearby provinces)."

Page 1, Line 32: Please define BTH at the first appearance. Abbreviations and acronyms are typically defined the first time they are used within the main text and then used throughout the remainder of the manuscript.

Response: We have added the full name of BTH and corrected the similar error in other places.

Page 1, Line 33-34: PMF was employed to apportion the source contribution to PM2.5. The sources of gaseous precursors are identified in this study?

Response: Actually, the sources of gaseous precursors were not identified in this study as their source apportionment is usually implemented by the method of emission inventory (Zhao et al., 2012), isotopic techniques (Zong et al., 2017) or air quality models, such as the Comprehensive Air Quality Model (CAMx) (Lu et al., 2016) and CMAQ model (Zhang et al., 2012). However, based on our dataset, we could not further apportion the secondary aerosols, therefore, we could not determine the exact contribution of the secondary transformation of gaseous precursors emitted by specific emission sources. That's what we want to investigate in the future.

Page 3, Lines 2-36: This part is not well written. The authors state that the studies on haze pollution in the BTH region have obtained fruitful and meaningful results. Please summarize the results on the chemical speciation, haze formation mechanism, emission sources, and influences of regional transport in North China. Meanwhile, the authors need to logically address why it is particularly important to do this study. The temporal and spatial characteristics can not be obtained based on single-site and short-term study. However, we still could get information from a lot of previous studies. What science questions need more research, particularly based on multiple-site and long-term study? Please clarify them.

Response: Thanks for the valuable suggestions to help us improve the instruction of this manuscript. Following these suggestions, first, we have further summarized the representative results related to the chemical speciation, haze formation mechanism, emission sources, influences of regional transport, as well as suggested mitigation strategies in North China, this part could be referred in Page 3, line 11−29 of the manuscript.

Second, we also logically address why it is particularly important to do this study, through summarizing the deficiency of previous literature researches and the existing scientific issues. Then, we have listed three aspects that why we need multiple-site and long-term research in Beijing-Tianjin-Hebei region (Page 3, line 32 to Page 4, line 4) .

The revised contents are added below:

" Extensive studies have been performed to investigate the formation mechanisms and emission sources of haze pollution in the BTH region and have obtained many valuable results (Du et al., 2014; Liu et al., 2016; Sun et al., 2013; Wang et al., 2014; Zhang et al., 2014; Zhao et al., 2013b). Massive anthropogenic emissions from diverse local sources, such as regional civil/industrial energy consumption, urban traffic, biomass burning and resuspended dust, and those transported from nearby provinces are widely regarded as the intrinsic reasons behind regional haze pollution events (Zhang et al., 2013; Zhao et al., 2012). Abnormal and unfavorable weather conditions also act as crucial factors in the formation of extensive and prolonged haze pollution events, such as the persistent haze event in January 2013 (Tao et al., 2014; Wang et al., 2014). In addition, many case studies, such as the winter regional haze events of 2010 (Zhao et al., 2013b) and 2013 (Sun et al., 2014; Wang et al., 2014), have also revealed that severe haze events are largely driven by the

high secondary production of sulfate, nitrate, ammonium and secondary organic aerosols (SOA), suggesting that aerosol chemistry plays a dominant role in haze evolution. Recent studies have reported a new efficient formation pathway for sulfate in the Beijing winter haze via reactive nitrogen chemistry in aerosol water during haze events (Cheng et al., 2016a; He et al., 2014; Wang et al., 2016, 2013). In terms of of haze mitigation strategies, Guo et al. (2014) suggested that regulatory controls of gaseous emissions for volatile organic compounds and nitrogen oxides from local transportation and sulfur dioxide from regional industrial sources are the keys to reducing the urban PM level in Beijing. However, these studies were often conducted at single sites (mostly in Beijing) and/or for short periods (specific haze events or a certain season); long-term multisite studies are scarce (Li et al., 2017; Shen et al., 2016; Zhang et al., 2013; Zhao et al., 2013c; Zong et al., 2016). In such studies, further questions are raised: first, due to the relatively few studies in Tianjin and Hebei, especially with respect to the source explorations of $PM_{2.5}$, we cannot fully understand the overall characteristics of the haze pollution in the BTH region; second, it is hard to directly compare the results between single-site studies to conduct a regional assessment as these studies covered different time periods and were conducted using different analytical approaches; third, the spatial and temporal variability of the $PM_{2.5}$ sources in this region have not been extensively investigated, particularly with respect to the evolutions of emission sources at different pollution levels and their spatial variability. The above imperfections can limit the understanding of the sources and evolution processes of haze pollution on a regional scale and complicate effective mitigation strategies."

Page 4, Lines 4-6: The authors state that they emphatically analyzed the chemical compositions and emission sources at different levels and the influence of air masses. Please highlight the new results or findings on them in the abstract.

Response: we have highlight it now in the abstract (Page 1, line 23–33), as follows:

" Secondary inorganic aerosol play a vital role in the haze process, with the exception of the spring haze of Shijiazhuang and Tianjin, where the dust source was crucial. In addition to secondary transformations, local direct emissions (coal combustion and motor vehicle exhaust) significantly contribute to the winter haze at the urban sites. Moreover, with the aggravation of haze pollution, the OC/EC mass ratio of $PM_{2.5}$ decreased considerably and the nitrate-rich secondary aerosol increased during all four seasons in Beijing, both of which indicate that local motor vehicle emissions significantly contribute to the severe haze episodes of Beijing. To assess the impacts of regional transport on haze pollution, the PMF results were further processed with backward trajectory clusters analysis, revealing that haze pollution usually occurred when air masses originating from polluted industrial regions in the south prevailed and is characterized by high $PM_{2.5}$ loadings with considerable contributions from secondary aerosol."

Page 4, Line 14: "These site reflect the atmospheric pollutions condition in this region"? Reword this sentence. Please clarify the sampling strategy.

Response: In this study, we selected four sampling sites in the Beijing-Tianjin-Hebei region, including three urban sites (Beijing, Tianjin and Shijiazhuang) and a regional background site (Xinglong). Beijing is the capital of China, and Tianjin is an economically developed municipality and is a coastal city. They have experienced rapid economic development and sharp increases in population, huge energy consumption and vehicle population, and thus induced a great amount of

pollutants; meanwhile, they are often suffered from the transported pollutants from heavily polluted areas (especially in the south direction). Therefore, Beijing and Tianjin have been reported as the most polluted megacities among 45 global megacities (Cheng et al., 2016b). As the capital of Hebei province, Shijiazhuang is a populated, industrialized and urbanized inland city. In Shijiazhuang, a great deal of coal are combusted for industrial processes and daily life, and the number of cars increases rapidly year by year, thus it is frequently reported as the most polluted city in the North China (Zhao et al., 2013a). Figure R1 shows the annual average concentrations of $PM_{2.5}$ and $PM_{10}$ of 2013 in major cities in Beijing-Tianjin-Hebei region (the data are provided by the National Environmental Bureau), from which we can see clearly that the PM concentrations were at a high level in Hebei province and extremely severe in Shijiazhuang.

The atmospheric pollution in these three cities is very serious, but has their own characteristics due to different energy and industrial structures. Therefore, they can represent the pollution characteristics of different urban types. In addition, for the aspect of specific selection of site location, we chose the sites that are affected by non-specific pollution sources while influenced by mixed emission sources, such as local motor vehicle emissions, coal combustion, road dust, industrial activities, cooking, transported pollutants, etc. These selected sites are considered to be representative of typical urban environments. Moreover, the comparative analysis of the city sites and the background site can reflect the contribution of anthropogenic sources to regional atmospheric pollution.

In addition, the sampling strategy has been briefly clarified in the manuscript in Page 4, line 19 to Page 5, line 4.

[Figure]

Figure R1 The annual average concentration of PM of 2013 in Beijing, Tianjin and major cities in Hebei province.

Page 4, Lines 27-28: Please clarify the distance between the filter sampling site and meteorological monitoring station in Tianjin, Shijiazhuang and Xinglong.
Response: The distance between the sampling site and meteorological monitoring station has been added now in Section 2.4. In addition, considering the nearby meteorological monitoring station of China Meteorological Administration (CMA) in Xinglong is still a few kilometers away from the sampling site and the data in April is missed, so we were actively seeking more appropriate meteorological data. Fortunately, we have gained the data from one meteorological monitoring station established by the Institute of Atmospheric Physics (IAP) of the Chinese Academy of Sciences (CAS) is right there, within 50 m of the sampling site in Xinglong. We conducted a

comparison between the two dataset and found significant differences existed, which may largely result from the long distance (a few kilometers) between the two meteorological monitoring stations. Therefore, we replaced the previous meteorological data gained from CMA with that from IAP to be more accurate, which could be referred in the Supplement Table S1.

Page 7, Line 11: The calculation of mineral dust was performed on the basis of six crustal element oxides. Why are Na, Mg, Zn not included in the Mineral dust? The calculations of Si, K and Fe are based on their ratios to Al in crustal dust. References should be added here.

Response: In this study, Na was not measured due to poor data quality, and the enrichment factor of Zn in $PM_{2.5}$ was much larger than 10, and exhibited much higher values on heavy pollution days (280–543 at the four sites) than that on clean days (73–201), indicating Zn was not primarily from natural sources and instead it came mainly from anthropogenic sources. Therefore, the calculation of mineral dust on the basis of six crustal element oxides (Al2O3, SiO2, CaO, $Fe_2O_3$, $MnO_2$ and $K_2O$) was referred to the literature by Christoforou et al. (2000) and Chow et al. (1994), and neither of the two calculation methods took Na, Mg and Zn into account. This calculation method has been applied in the study of $PM_{2.5}$ in Beijing by He et al. (2001). In addition, considering the absence of measuring Si due to the material of filter (Quartz membrane), and some anthropogenic sources for K and Fe, the calculations of Si, K and Fe are based on their ratios to Al in crustal dust, the references of these ratio have been added in the revised paper, please refer to Section 2.3.1 in Page 7.

Page 8, Lines 26-29: The authors state that the cycles of haze episodes are primarily driven by fluctuations in meteorological conditions such as wind speed, relative humidity, air temperature, atmospheric stability, the height of the planetary boundary layer and air mass origins. Please show the temporal trend of these meteorological parameters in 4-7 days here or in Supporting information. In my opinion, these parameters have more effect on the diurnal variation of air pollutants rather than the 4-7 days' cycle.

Response: Actually, we quite agree that meteorological parameters have more effect on the diurnal variation of air pollutants than the periodic cycle of a few days. However, the meteorological conditions still represent one of the most critical parameters in regulating the cycles of pollution episodes in Beijing by influencing the regional transport, accumulation and scavenging of pollutants. Generally, local emissions show a steadier pattern, the sudden and apparent sharp increase in $PM_{2.5}$ concentration recorded in Beijing represents rapid recovery from an interruption to the continuous pollution accumulation over the region, rather than purely local chemical production or enhanced emissions (Zheng et al., 2015). Therefore, meteorological condition plays an external role in the occurrence of pollution episodes. Then, under unfavorable synoptic meteorological conditions, local aerosol chemical processes, most likely heterogeneous reactions, would cause the rapid growth of particle mass and lead to severe $PM_{2.5}$ development.

The PM mass concentration is primarily sensitive to the fluctuations in wind speed and planetary boundary layer (PBL) height (Yang et al., 2015). For example, Guo et al. (2014) observed that the wind variation correlates well with the $PM_{2.5}$ events in Beijing. During the transition and polluted periods, the wind direction and wind speed play an important role. In Beijing, the wind shifted from northerly to southerly with a considerably decreasing speed could result in air masses from the more populated southern industrial regions and a stagnant condition.

The 4-7 days' cycle of haze episodes in Beijing observed by Guo et al. (2014) has been also recorded by Zheng et al. (2015) and Jia et al. (2008). However, in this study, we did not delve into the periodic cycle of haze episodes, considering it is not the focus of this study and the variation of meteorological parameters and $PM_{2.5}$ mass concentration cannot be well matched due to that $PM_{2.5}$ mass concentrations in this study are the average of 11.5 hours.

Page 9, Lines 26-35: The extreme pollution events on Oct 5-11 and Jan 2-6, Jan 11-16 are worth a in-depth discussion.

Response: In this Section 3.1.2, we have briefly described the extreme pollution events during the observation, and in Section 3.4.1 we have added a in-depth discussion on the pollution events in summer, winter and spring, through analyzing the variation of chemical components during the specific haze episode in different seasons, the degree of secondary formation of sulfur and nitrogen, and the possible formation and development mechanism of haze pollution (Page 21, line 7 to Page 22, line 30). Considering we have missed so many samples due to imperfection of sampling, we have not deeply discussed the extreme pollution events on Oct 5–11, but cited the results of by Yang et al. (2015), who have also recorded this episode and other haze episodes in October 2014, and have analyzed these episodes in detail.

[revised manuscript text omitted]

Page 10, Line 9: Shijiazhuang recorded high relative humidity and low wind speeds. It is usual or not from the historical record. Please clarify.

Response: In this study, higher relative humidity and lower wind speeds were recorded in Shijiazhuang than that in Tianjin and Beijing. This is usual from the historical record after statistical analysis of the data gained from the airport sites in the website of https://www.wunderground.com/ from 2004 to 2014, the monthly average values of relative humidity and wind speed are shown in Fig R2. It could be clearly seen lower wind speed in Shijiazhuang than that in Beijing and Tianjin, the average wind speed during 2004–2014 in Shijiazhuang was 2.2 m/s, while 3.0 m/s in Beijing and 3.3 m/s in Tianjin. The differences in relative humidity are not significant between these three sites, however, from the average values during 2004–2014, the relative humidity was 61% in Shijiazhuang and Tianjin, and 56% in Beijing.

[Figure]

Figure R2 Monthly mean variations of relative humidity (RH) and wind speed from 2004 to 2014 at the airport sites in Beijing, Tianjin and Shijiazhuang

Page 11, Lines 10-21: Please provide the standard deviation of the ratio of NO3-/SO42- in four sampling sites. The data reported here is annual average value? Please clarify.

Response: Even though the annual average value of $NO_3^-/SO_4^{2-}$ mass ratio is more reasonable, considering the comparison with previous studies, the mass ratio of $NO_3^-/SO_4^{2-}$ reported here is calculated based on the annual average concentration of $NO_3^-$ and $SO_4^{2-}$ as did by Zhao et al. (2013b). The results of the two calculation methods are very close in this study. Therefore, there is no standard deviation for the ratio of $NO_3^-/SO_4^{2-}$. In order to avoid such ambiguity, we have clarified this in the revised paper.

Page 12, Lines 16-17: The authors state that strengthened burning activities may occur at night because of the higher night/day ratios of EC and Cl- than that of PM2.5. Actually the photochemical reaction of secondary species and the boundary layer variation also could result in the higher night/day ratio of primary PM. Please provide more evidences to support this statement.

Response: Thank the referee for pointing out this. The statement of "strengthened burning activities may occur at night, as the night/day ratios of EC and Cl⁻, ... , were higher than those of $PM_{2.5}$." was only based on speculation, and we ignored the important factor that the photochemical reaction of secondary species and the boundary layer variation also could result in the higher night/day ratio of primary PM. Therefore, we have deleted this statement now.

Page 12, Lines 29-31: Please provide the meteorological parameters at day and night in the four sites in the supporting information to support this statement.

Response: It has been added in Table S2 in the Supplement, and the meteorological parameters at different pollution level in each season have also added in Table S2.

Page 15: I still think the analysis on the whole pollution processes particularly the extreme pollution events could get more information on haze formation mechanism than that of the different pollution levels.

Response: Thanks for this good suggestion. We agree that the study of a whole pollution processes could get more information about the haze formation mechanism and evolution process during specific haze pollution event. Therefore, even though we mainly aimed to reveal the average state of haze pollution in the BTH region in this study, the analysis on the whole severe pollution processes is still necessary. Therefore, in addition to the evolution of chemical components at different pollution levels during the entire observation period from 2014–2015 and in each season, we added the analysis on the whole pollution processes in summer (Jun 29 to Jul 8), winter (Jan 11 to 16) and spring (Mar 23 to Apr 1). The variation of chemical components, the sulfur oxidation ratio and nitrogen oxidation ratio are presented in Figure 8, and the time series of meteorological parameters during the three periods are also given in Figure S6 in the Supplement. The added contents can be referred to from Page 21, line 7 to Page 22, line 30).

Page 15, Line 32: The OC/EC ratio increases with the increasing development of haze pollution? It is different from the statement in Lines 19-20. Is there any study on the secondary organic

carbon in the wintertime of Beijing and Shijiazhuang? Maybe the sources affect the OC/EC ratio in different pollution levels. Again, the discussion on the specific pollution process will avoid this bias associated with emission sources.

Response: We are sorry for our carelessness. Actually, the OC/EC mass ratio in Beijing decreased with the increasing development of haze pollution. The wrong presentation has been revised now.

The formation of secondary organic carbon and the source changes at different pollution levels both can affect the OC/EC ratio. There are a lot of studies on the secondary organic carbon in the wintertime of Beijing, but very few in Shijiazhuang. Therefore, the speculation about the reason for the increase of OC/EC ratio in Shijiazhuang lacks accurate evidence, and thus we decided to delete related comments in the manuscript.

Page 16, Lines 22-25: I strongly suggest to discuss the differences between the spring haze and winter haze. The formation mechanism may be totally different.

Response: According to this suggestion, we conducted the discussion on the differences between the spring haze and winter haze. Just as the referee predicted, the formation mechanisms are totally different, the winter haze was mainly formed by local processes (local direct emissions and secondary transformation), while dust drove the formation of spring haze, especially at Shijiazhuang and Tianjin. The related discussion are presented as below (Page 23, line 21 to Page 24, line 29):

" In contrast to background site, the emission sources and generation mechanisms of haze pollution were more complex at the urban sites, especially in winter, as the primary emissions, such as the motor vehicle exhaust, coal combustion and industrial processes were also the main sources of heavy pollution in winter. As the main fuel for winter heating in northern China, the contribution of coal combustion to $PM_{2.5}$ mainly occurred in winter and was key to the heavy pollution in winter in urban areas, as the contribution increases with increasing pollution levels. In Tianjin and Shijiazhuang, this process contributed nearly 30% to the $PM_{2.5}$ on heavy pollution days and contributed even more when considering the secondary formation of gaseous precursors emitted by coal combustion. Moreover, the primary emissions of motor vehicles also exerted a remarkable impact on the winter haze pollution, accounting for 26.1% of $PM_{2.5}$ on heavy pollution days in Beijing, and accounting for more when the secondary conversion of gaseous pollutants in vehicle exhaust are considered, as nitrate-rich secondary aerosol increased from 2.9% on clean days to 19.5% on heavy pollution days during the winter period. On hazy days, low visibility could aggravate urban traffic congestion during rush hour, thus causing more pollutants to be emitted by motor vehicles operating in these conditions (Zhang et al., 2011). Therefore, the winter haze was mainly formed by local processes (local direct emissions and secondary transformation).

In spring, the effect of dust source was highlighted at the urban sites. Most notably, in Shijiazhuang, the mineral dust source significantly contributed to the aerosol pollution process, as its contribution to $PM_{2.5}$ continuously increased from 9.7% on clean days to 18.6% on moderate pollution days to 22.9% on heavy pollution days. However, along with the increase in pollution levels, the ratio of local road dust source to the overall dust source decreased from 81.6% (on clean days) to 50% (on pollution days), thus reflecting the significant impact of the long-range transport of the northwest dust on the spring aerosol pollution in Shijiazhuang. Differing from the increase of relative humidity during haze episodes of the other seasons, the relative humidity decreased continuously from 64% (clean days) to 47% (moderate pollution days) and to 40%

(heavy pollution days) in spring in Shijiazhuang (Table S2). Therefore, the heterogeneous reactions promoted by enhanced water vapor may not be the spring haze formation mechanism in Shijiazhuang, as the contribution of secondary inorganic aerosol decreased remarkably during the haze episode (Fig. 9 and Fig. S5). In contrast, the wind speed and the contribution of the dust source significantly increased during the spring haze period in Shijiazhuang. A similar but much milder pattern (except in terms of wind speed) was also recorded in Tianjin. Therefore, dust pollution, which was mainly the result of long-range transported soil dust and local road dust, contributed to the spring haze in Shijiazhuang and Tianjin."

Page 21, Lines 19-page 22, Line 19: The authors should consider incorporating this discussion into the other sections of the manuscript so that comparisons are made when results are discussed.
Response: Thanks for the good suggestion. We have merged this part (now Section 3.4.2) with the evolution of chemical components (now Section 3.4.1) into Section 3.4. Please refer to the revised manuscript.

**Chemical characterization and source identification of PM$_{2.5}$ at multiple sites in the Beijing-Tianjin-Hebei region, China**

**Xiaojuan Huang[1,2], Zirui Liu[1,3*], Jingyun Liu[1], Bo Hu[1], Tianxue Wen[1], Guiqian Tang[1], Junke Zhang[1], Fangkun Wu[1], Dongsheng Ji[1], Lili Wang[1], Yuesi Wang[1,3*]**

[1]State Key Laboratory of Atmospheric Boundary Layer Physics and Atmospheric Chemistry (LAPC), Institute of Atmospheric Physics, Chinese Academy of Sciences, Beijing, China

[2]Plateau Atmosphere and Environment Key Laboratory of Sichuan Province, School of Atmospheric Sciences, Chengdu University of Information Technology, Chengdu, China

[3]Center for Excellence in Regional Atmospheric Environment, Institute of Urban Environment, Chinese Academy of Sciences, Xiamen, China

[*]Corresponding author: Zirui Liu (liuzirui@mail.iap.ac.cn); Yuesi Wang (wys@mail.iap.ac.cn)

**ABSTRACT:** The simultaneous observation and analysis of atmospheric fine particles (PM$_{2.5}$) on a regional scale is an important approach to develop control strategies for haze pollution. In this study, samples of filtered PM$_{2.5}$ were collected simultaneously at three urban sites (Beijing, Tianjin, and Shijiazhuang) and at a regional background site (Xinglong) in the Beijing-Tianjin-Hebei (BTH) region from June 2014 to April 2015. The PM$_{2.5}$ at the four sites mainly comprised organic matter, secondary inorganic ions (sulfate, nitrate and ammonium) and mineral dust. Positive matrix factorization (PMF) demonstrated that, on an annual basis, secondary inorganic aerosol was the largest PM$_{2.5}$ source in this region, accounting for 29.2–40.5% of the PM$_{2.5}$ mass at the urban sites; the second largest PM$_{2.5}$ source was primary emission of motor vehicle exhaust in Beijing (24.9%), whereas coal combustion in Tianjin (15.2%) and Shijiazhuang (17.3%), particularly in winter. Secondary inorganic aerosol play 
[revised manuscript text omitted]

Table S2 The meteorological parameters under specific conditions during the sampling periods

| | | Temperature (℃) | | | | Relative humidity (%) | | | | Wind speed (m/s) | | | |
|---|---|---|---|---|---|---|---|---|---|---|---|---|---|
| | | Summer | Autumn | Winter | Spring | Summer | Autumn | Winter | Spring | Summer | Autumn | Winter | Spring |
| Beijing | Annual | 28.0 | 18.0 | 1.5 | 14.0 | 51 | 59 | 32 | 35 | 1.0 | 0.8 | 1.5 | 1.7 |
| | C[a] | 28.3 | 18.6 | 1.5 | 12.1 | 44 | 46 | 21 | 32 | 1.2 | 1.2 | 2.3 | 2.3 |
| | MP[b] | 27.4 | 18.7 | 2.3 | 14.2 | 57 | 69 | 34 | 33 | 0.9 | 0.6 | 1.4 | 1.6 |
| | HP[c] | 27.7 | 17.5 | 1.1 | 16.4 | 67 | 69 | 41 | 42 | 0.7 | 0.6 | 0.9 | 1.2 |
| | D[d] | 30.3 | 19.8 | 2.8 | 16.0 | 43 | 51 | 28 | 27 | 1.4 | 1.0 | 1.7 | 2.1 |
| | N[e] | 25.8 | 16.2 | 0.1 | 11.2 | 60 | 67 | 37 | 43 | 0.7 | 0.6 | 1.3 | 1.3 |
| Tianjin | Annual | 27.7 | 18.4 | 1.1 | 13.0 | 54 | 60 | 41 | 41 | 1.5 | 1.3 | 1.4 | 1.8 |
| | C | 28.6 | 18.9 | 1.5 | 11.2 | 47 | 61 | 31 | 39 | 1.6 | 1.4 | 1.8 | 2.1 |
| | MP | 27.2 | 19.1 | 1.1 | 13.3 | 59 | 61 | 40 | 44 | 1.4 | 1.1 | 1.6 | 1.8 |
| | HP | 28.4 | 16.9 | 1.7 | 13.4 | 66 | 53 | 46 | 40 | 1.2 | 1.2 | 1.1 | 1.5 |
| | D | 29.7 | 20.1 | 2.6 | 15.2 | 47 | 53 | 35 | 34 | 1.8 | 1.5 | 1.7 | 2.1 |
| | N | 25.7 | 16.6 | -0.4 | 10.9 | 62 | 69 | 45 | 48 | 1.2 | 1.0 | 1.2 | 1.5 |
| Shijiazhuang | Annual | 26.9 | 17.9 | 1.0 | 13.8 | 63 | 75 | 41 | 46 | 1.2 | 0.9 | 1.0 | 1.5 |
| | C | 28.6 | 16.3 | 1.9 | 9.1 | 50 | 64 | 21 | 64 | 1.3 | 1.1 | 1.3 | 1.2 |
| | MP | 26.6 | 18.9 | 1.6 | 13.0 | 66 | 78 | 35 | 47 | 1.2 | 0.9 | 1.4 | 1.5 |
| | HP | 26.2 | 17.7 | 0.6 | 15.5 | 71 | 78 | 46 | 40 | 1.1 | 0.7 | 0.9 | 1.6 |
| | D | 29.2 | 19.6 | 3.0 | 16.0 | 54 | 68 | 36 | 39 | 1.4 | 1.1 | 1.2 | 1.9 |
| | N | 24.8 | 16.2 | -1.0 | 11.5 | 70 | 83 | 47 | 53 | 1.0 | 0.7 | 0.8 | 1.2 |
| Xinglong | Annual | 20.6 | 12.1 | -4.8 | 7.0 | 67 | 66 | 40 | 44 | 2.3 | 2.5 | 2.7 | 3.3 |
| | C | 20.6 | 12.1 | -5.0 | 6.5 | 61 | 58 | 35 | 30 | 2.1 | 2.6 | 2.9 | 3.4 |
| | MP | 19.7 | 14.0 | -3.1 | 8.9 | 80 | 68 | 42 | 52 | 2.5 | 3.3 | 2.6 | 3.4 |
| | HP | 21.5 | 13.8 | -4.9 | 8.9 | 85 | 74 | 79 | 66 | 2.7 | 2.4 | 1.4 | 3.6 |
| | D | 21.7 | 13.2 | -3.7 | 8.3 | 64 | 62 | 36 | 38 | 2.4 | 2.7 | 2.7 | 3.6 |
| | N | 19.4 | 10.9 | -5.8 | 5.5 | 70 | 71 | 43 | 50 | 2.1 | 2.3 | 2.7 | 3.1 |

[a] Clean days (PM$_{2.5}$＜75 μg/m$^3$); [b] Moderate pollution days (75≤PM$_{2.5}$＜150 μg/m$^3$); [c] Heavy pollution days (PM$_{2.5}$≥150 μg/m$^3$);

[d] Daytime; [e] Nighttime

**Table S3 Comparisons of the PM$_{2.5}$ source apportionment results from this study and other recent studies**

| Reference | Wu et al. (2014) | | Liu et al. (2017) | | EPB[a] | | | | This study | | | |
|---|---|---|---|---|---|---|---|---|---|---|---|---|
| City | BJ[b] | | TJ[c] | | | BJ | TJ | SJZ[d] | | BJ | TJ | SJZ |
| Time | 2010 | | 06-08/2015 | | 2013-2014 | | | | 2014-2015 | | | |
| Method | PMF | | PMF | | multiple models | | | | PMF | | | |
| Sources and contributions (%) | traffic emissions | 12.0 | vehicle exhaust | 25.4 | vehicle exhaust | 19.9–22.4 | 13.2–15.6 | 10.5–11.6 | motor vehicle exhaust | 24.9 | 15.2 | 17.3 |
| | coal combustion | 22.0 | coal combustion | 16.5 | coal combustion | 14.3–16.1 | 17.8–21.1 | 20.0–21.9 | coal combustion | 5.6 | 12.4 | 15.5 |
| | secondary sulfate/nitrate | 30.2 | secondary sources | 26.1 | | | | | secondary inorganic | 40.5 | 29.2 | 36.4 |
| | dust/soil | 12.4 | crustal dust | 13.2 | soil dust | 9.2–10.3 | 19.8–23.4 | 15.8–17.3 | mineral dust | 8.6 | 11.7 | 8.5 |
| | metallurgical emission | 0.4 | | | | | | | oil refining /metal smelting source | | 2.8 | 0.7 |
| | industry | 6.9 | | | industry | 11.6–13.0 | 11.2–13.3 | 17.6–19.4 | industrial process | 3.2 | 8.9 | 6.1 |
| | secondary organic aerosol | 9.9 | | | regional transport | 28–36 | 22–34 | 23–30 | | | | |
| | | | biomass burning | 10.2 | other sources[e] | 9.0–10.2 | 4.0–4.7 | 6.2–6.8 | biomass burning | 4.5 | 5.3 | 2.8 |

[a] Beijing Municipal Environmental Protection Bureau (http://www.bjepb.gov.cn/), Tianjin Municipal Environmental Protection Bureau (http://www.tjhb.gov.cn/) and Shijiazhuang Municipal Environmental Protection Bureau (http://www.sjzhb.gov.cn/)

[b] Beijing; c Tianjin; d Shijiazhuang

[e] Including emissions from biomass burning, cooking, and agricultural, etc.

[Figure]

Figure S1. Gravimetric PM$_{2.5}$ versus reconstructed PM$_{2.5}$ mass concentrations from Beijing (a), Tianjin (b), Shijiazhuang (c) and Xinglong (d). "n" represents the sample quantity at each site.

[Figure]

Figure S2. PMF factor/source profiles for the PM$_{2.5}$ samples throughout the entire study period in Beijing in terms of concentrations (μg/m$^3$) and percentages (%)

[Figure]

Figure S3. PMF factor/source profiles for the PM$_{2.5}$ samples throughout the entire study period in Tianjin in terms of concentrations (μg/m$^3$) and percentages (%)

[Figure]

Figure S4. PMF factor/source profiles for PM$_{2.5}$ samples throughout the entire study period in Shijiazhuang in terms of concentrations (μg/m$^3$) and percentages (%)

[Figure]

Figure S5. The mass fraction of each aerosol chemical species at different pollution levels throughout the entire study period. C, MP, and HP represent clean days (PM$_{2.5}$＜75 μg/m$^3$), moderate pollution days (75≤PM$_{2.5}$＜150 μg/m$^3$) and heavy pollution days (PM$_{2.5}$≥150 μg/m$^3$), respectively. "%" represents the proportion of filter sample quantity at each pollution level to the total samples.

[Figure]

Figure S6. Time series variations of the meteorological parameters from Beijing during the specific pollution periods in summer (a), winter (b) and spring (c).

**Supplementary text for describing the day-night variations of PM$_{2.5}$ and major chemical components**

The analysis of the day-night variations indicates that the differences in the PM$_{2.5}$ annual average concentrations during the day and those during the night were significant at the urban sites, where the values were 8–19% higher at night than those during the day, while negligible differences were found on the annual scale in Xinglong (Fig. S7). This obvious day-night variation of the PM$_{2.5}$ concentrations in urban areas was probably due to the apparent changes in the height of the mixing layer between day and night (Zhao et al., 2009). However, in Xinglong, the dominant source was from the regional or long-range transport, with fewer contributions from local emissions; thus, the nocturnal stable boundary layer could have reduced the quantity of transmissions from the outside. The chemical compositions also recorded obvious day-night variations, as the mass ratio of NO$_3^-$/SO$_4^{2-}$ recorded higher values at night (0.99–1.39) than during the day (0.81–1.13), which is consistent with the similar results obtained by Sun et al. (2016) in Xianghe, which is located approximately 50 km southeast of Beijing. Such day-night variations indicate the important role of the gas-phase photochemical production of sulfate during the day while the facilitated gas-to-particle partitioning of semi-volatile nitrate is associated with the low temperatures (Sun et al., 2016) and effective hydrolysis of N$_2$O$_5$ at night, which is a major source of nitric acid in the urban atmosphere during the night and is more efficient on wet surfaces (Zhang et al., 2015). In addition, the relatively static and stable meteorological conditions at night resulted in obviously lower fractions of mineral dust (11.3–17.0%, except for in Tianjin), than those recorded during the day (18.3–24.3%).

[Figure]

Figure S7. Day-night variations of PM$_{2.5}$ and major chemical components, based on annual data

---

## Author Comment (AC2) · 15 Sep 2017

The comment was uploaded in the form of a supplement:
https://www.atmos-chem-phys-discuss.net/acp-2017-446/acp-2017-446-AC2-supplement.pdf